# DIFFERENTIALLY PRIVATE DATASET CONDENSATION

## ABSTRACT

Recent work in ICML'22 builds a theoretical connection between dataset conden-
sation (DC) and differential privacy (DP) and claims that DC can provide privacy
protection for free. However, the connection is problematic because of two con-
troversial assumptions. In this paper, we revisit the ICML'22 work and elucidate
the issues in the two controversial assumptions. To correctly connect DC and DP,
we propose two differentially private dataset condensation (DPDC) algorithms—
LDPDC and NDPDC. Through extensive evaluations, we demonstrate that LD-
PDC has comparable performance to recent DP generative methods despite its
simplicity. NDPDC provides acceptable DP guarantees with a mild utility loss,
compared to distribution matching (DM). Additionally, NDPDC allows a flexible
trade-off between the synthetic data utility and DP budget.

## 1 INTRODUCTION

Although deep learning has pushed forward the frontiers of many applications in the past decade,
it still needs to overcome some challenges for broader academic and commercial use (Dilmegani,
2022). One challenge is the costly process of algorithm design and practical implementation in deep
learning, which typically requires inspection and evaluation by training many models on ample data.
The growing privacy concern is another challenge. Due to the privacy concern, an increasing number
of customers are reluctant to provide their data for the academia or industry to train deep learning
models, and some regulations are further created to restrict access to sensitive data.

Recently, dataset condensation (DC) has emerged as a potential technique to address the two chal-
lenges with one shot (Zhao et al., 2021; Zhao & Bilen, 2021b). The main objective of dataset
condensation is to condense the original dataset into a small synthetic dataset while maintaining
the synthetic data utility to the greatest extent for training deep learning models. For the first chal-
lenge, both academia and industry can save computation and storage costs if using DC-generated
small synthetic datasets to develop their algorithms and debug their implementations. In terms of
the second challenge, since the DC-generated synthetic data may not exist in the real world, sharing
DC-generated data seems less risky than sharing the original data.

Nevertheless, DC-generated data may memorize a fair amount of sensitive information during the
optimization process on the original data. In other words, sharing DC-generated data still exposes the
data owners to privacy risk. Moreover, this privacy risk is unknown since the prior literature on DC
has not proved any rigorous connection between DC and privacy. Although an ICML'22 outstanding
paper (Dong et al., 2022) proved a proposition to support the claim that DC can provide certain
differential privacy (DP) guarantee for free, the proof is problematic because of two controversial
assumptions, as discussed in Section 3.1.

To correctly connect DC and DP for bounding the privacy risk of DC, we propose two differentially
private dataset condensation (DPDC) algorithms—LDPDC and NDPDC. LDPDC (Algorithm 1)
adds random noise to the sum of randomly sampled original data and then divides the randomized
sum by the fixed expected sample size to construct the synthetic data. Based on the framework of
Rényi Differential Privacy (RDP) (Mironov, 2017; Mironov et al., 2019), we prove Theorem 3.1 to
bound the privacy risk of LDPDC. NDPDC (Algorithm 2) optimizes randomly initialized synthetic
data by matching the norm-clipped representations of the synthetic data and the randomized norm-
clipped representations of the original data. For NDPDC, we prove Theorem 3.2 bound to its privacy

---

We note that, in this paper, performance specifically refers to the performance in producing high-utility
synthetic data for data-efficient model training.

risk. The potential benefits brought by DPDC algorithms include (i) reducing the cost of data storage and model training; (ii) mitigating the privacy concerns from data owners; (iii) providing DP guarantees for the hyperparameter fine-tuning process and the trained models on the DPDC-generated synthetic datasets, due to the post-processing property.

We conduct extensive experiments to evaluate our DPDC algorithms on multiple datasets, including MNIST, FMNIST, CIFAR10, and CelebA. We mainly compare our DPDC algorithms with a non-private dataset condensation method, *i.e.*, distribution matching (Zhao & Bilen, 2021a), and two recent differentially private generative methods, *i.e.*, DP-MERF and DP-Sinkhorn (Cao et al., 2021; Harder et al., 2021). We demonstrate that (i) LDPDC shows comparable performance to DP-MERF and DP-Sinkhorn despite its simplicity; (ii) NDPDC can provide DP protection with a mild utility loss, compared to distribution matching; (iii) NDPDC allows a flexible trade-off between privacy and utility and can use low DP budgets to achieve better performance than DP-MERF and DP-Sinkhorn.

## 2 BACKGROUND AND RELATED WORK

### 2.1 DATASET CONDENSATION

We denote a data sample by $x$ and its label by $y$. In this paper, we mainly study classification problems, where $f_{\theta}(\cdot)$ refers to the model with parameters $\theta$. $\ell(f_{\theta}(x), y)$ refers to the cross-entropy between the model output $f_{\theta}(x)$ and the label $y$. Let $\mathcal{T}$ and $\mathcal{S}$ denote the original dataset and the synthetic dataset, respectively, then we can formulate the dataset condensation problem as

$$\arg\min_{\mathcal{S}} \mathbb{E}_{(x,y)\sim\mathcal{T}} \ell(f_{\theta(\mathcal{S})}(x), y), \quad \text{where } \theta(\mathcal{S}) = \arg\min_{\theta} \mathbb{E}_{(x,y)\sim\mathcal{S}} \ell(f_{\theta}(x), y), |\mathcal{S}| \ll |\mathcal{T}| \quad (1)$$

An intuitive method to solve the above objective is meta-learning (Wang et al., 2018), with an inner optimization step to update $\theta$ and a outer optimization step to update $\mathcal{S}$. However, this intuitive method needs a lot of cost for implicitly using second-order terms in the outer optimization step. Nguyen et al. (2020) considered the classification task as a ridge regression problem and derived an algorithm called kernel inducing points (KIP) to simplify the meta-learning process. Gradient matching is an alternative method (Zhao et al., 2021) for dataset condensation, which minimizes a matching loss between the model gradients on the original and synthetic data, *i.e.*,

$$\min_{\mathcal{S}} \mathbb{E}_{\theta_0 \sim \mathbb{P}_{\theta_0}} \left[ \sum_{i=1}^{I-1} \Pi_M (\nabla_{\theta} \mathcal{L}^{\mathcal{S}}(\theta_i), \nabla_{\theta} \mathcal{L}^{\mathcal{T}}(\theta_i)) \right]. \quad (2)$$

$\Pi_M$ refers to the matching loss in (Zhao et al., 2021); $\nabla_{\theta} \mathcal{L}^{\mathcal{S}}(\theta_i)$ and $\nabla_{\theta} \mathcal{L}^{\mathcal{T}}(\theta_i)$ refer to the model gradients on the synthetic and original data, respectively; $\theta_i$ is updated on the synthetic data to obtain $\theta_{i+1}$. To boost the performance, Zhao & Bilen (2021b) further applied differentiable Siamese augmentation $\mathcal{A}_w(\cdot)$ with parameters $w$ to the original data samples and the synthetic data samples. Recently, Zhao & Bilen (2021a) proposed to match the feature distributions of the original and synthetic data for dataset condensation. Zhao & Bilen (2021a) used an empirical estimate of maximum mean discrepancy (MMD) as the matching loss, *i.e.*,

$$\mathbb{E}_{\theta \sim P_{\theta}} \left\| \frac{1}{|\mathcal{T}|} \sum_{i=1}^{|\mathcal{T}|} \Phi_{\theta}(\mathcal{A}_w(x_i)) - \frac{1}{|\mathcal{S}|} \sum_{i=1}^{|\mathcal{S}|} \Phi_{\theta}(\mathcal{A}_w(x_i)) \right\|_2^2, \quad (3)$$

where $\Phi_{\theta}(\cdot)$ is the feature extractor, and $P_{\theta}$ is a parameter distribution. With the help of differentiable data augmentation (Zhao & Bilen, 2021b), the distribution matching method (DM) (Zhao & Bilen, 2021a) achieves the state-of-the-art performance on dataset condensation.

### 2.2 DIFFERENTIAL PRIVACY

Differential Privacy (DP) (Dwork et al., 2006) is the most widely-used mathematical definition of privacy, so we first introduce the definition of DP in the following.

**Definition 2.1 (Differential Privacy (DP))** *For two adjacent datasets $D$ and $D'$, and every possible output set $\mathcal{O}$, if a randomized mechanism $\mathcal{M}$ satisfies $\mathbb{P}[\mathcal{M}(D) \in \mathcal{O}] \leq e^{\epsilon} \mathbb{P}[\mathcal{M}(D') \in \mathcal{O}] + \delta$, then $\mathcal{M}$ obeys $(\epsilon, \delta)$-DP.*

Based on the framework of DP, Abadi et al. (2016) developed DP-SGD with a moments accountant for learning differentially private models.

We also introduce the concept of Rényi Differential Privacy (RDP), as RDP gives us a unified view of pure DP and $(\epsilon, \delta)$-DP, graceful composition bounds, and tighter bounds for the (sub)sampled Gaussian mechanism (Definition A.1) (Wang et al., 2019; Mironov et al., 2019). Due to the benefits of RDP, Meta's Opacus library (Yousefpour et al., 2021) also relies on (Mironov et al., 2019) for DP analysis. We begin the brief introduction of RDP with two basic definitions:

**Definition 2.2 (Rényi Divergence (Rényi et al., 1961))** *Let $P$ and $Q$ be two distributions on $\mathcal{Z}$ over the same probability space, the Rènyi divergence between $P$ and $Q$ is*

$$\mathcal{D}_\alpha(P\|Q) \triangleq \frac{1}{\alpha - 1} \ln \int_{\mathcal{Z}} q(\boldsymbol{z})(\frac{p(\boldsymbol{z})}{q(\boldsymbol{z})})^\alpha d\boldsymbol{z}, \tag{4}$$

*where $p(\boldsymbol{z})$ and $q(\boldsymbol{z})$ are the respective probability density functions of $P$ and $Q$. Without ambiguity, $\mathcal{D}_\alpha(P\|Q)$ can also be written as $\mathcal{D}_\alpha(p(\boldsymbol{z})\|q(\boldsymbol{z}))$.*

**Definition 2.3 (Rényi Differential Privacy (RDP) (Mironov, 2017))** *For two adjacent datasets $D$ and $D'$, if a randomized mechanism $\mathcal{M}$ satisfies $\mathcal{D}_\alpha(\mathcal{M}(D)\|\mathcal{M}(D')) \leq \epsilon \ (\alpha > 1)$, then $\mathcal{M}$ obeys $(\alpha, \epsilon)$-RDP, where $\mathcal{D}_\alpha$ refers to Rényi divergence.*

We can easily connect RDP and DP by Lemma 2.1 and Corollary 2.1.

**Lemma 2.1 (RDP to DP Conversion (Balle et al., 2020))** *If a randomized mechanism $\mathcal{M}$ guarantees $(\alpha, \epsilon)$-RDP, then it also obeys $(\epsilon + \log((\alpha - 1)/\alpha) - (\log \delta + \log \alpha)/(\alpha - 1), \delta)$-DP.*

In Appendix A (Page 15), we show that Lemma 2.1 is tighter than Mironov (2017)'s conversion law.

**Corollary 2.1** *According to Lemma 2.1, if a mechanism $\mathcal{M}$ obeys $(\alpha, \epsilon(\alpha))$-RDP for $\alpha > 1$, then $\mathcal{M}$ obeys $(\min_{\alpha>1}(\epsilon(\alpha) + \log((\alpha - 1)/\alpha) - (\log \delta + \log \alpha)/(\alpha - 1)), \delta)$-DP.*

One main advantage of RDP is that RDP allows a graceful composition of the privacy budgets spent by multiple randomized mechanisms, as illustrated in Lemma 2.2.

**Lemma 2.2 (RDP Composition (Mironov, 2017))** *If $\mathcal{M}_1$ is $(\alpha, \epsilon_1)$-RDP, $\mathcal{M}_2$ is $(\alpha, \epsilon_2)$-RDP, then their composition obeys $(\alpha, \epsilon_1 + \epsilon_2)$-RDP.*

Furthermore, RDP allows a graceful parallel composition, as shown in Lemma 2.3.

**Lemma 2.3 (Parallel Composition (Chaudhuri et al., 2019))** *If two datasets $D_1$ and $D_2$ are disjoint ($D_1 \cap D_2 = \emptyset$), and $\mathcal{M}_1$ is $(\alpha, \epsilon_1)$-RDP, $\mathcal{M}_2$ is $(\alpha, \epsilon_2)$-RDP, then the combined release $(\mathcal{M}_1(D_1), \mathcal{M}_2(D_2))$ obeys $(\alpha, \max(\epsilon_1, \epsilon_2))$-RDP for $D_1 \cup D_2$.*

In Appendix E, we discuss more related work on differential privacy and generative methods.

## 3 DATASET CONDENSATION AND DIFFERENTIAL PRIVACY

### 3.1 REVISIT THE ICML'22 WORK (DONG ET AL., 2022)

Dong et al. (2022) first established a connection between DC and DP by Proposition 4.10 in their paper, which is unfortunately proved upon two controversial assumptions. One explicit controversial assumption is regarding the posterior model parameter distribution, as stated in Assumption 3.1.

**Assumption 3.1 (Assumption 4.8 in (Dong et al., 2022))** *Given the training dataset $\mathcal{S}$ and loss function $\ell$, the distribution of the model parameters is*

$$\mathbb{P}(\boldsymbol{\theta}|\mathcal{S}) = \frac{1}{Z_\mathcal{S}} \exp(-\sum_{i=1}^{|\mathcal{S}|} \ell(\boldsymbol{f_\theta}(\boldsymbol{s}_i), y_i)) \tag{5}$$

Although (Dong et al., 2022) presents the assumption on $\mathcal{T}$, it actually uses the assumption on the synthetic dataset $\mathcal{S}$ in the proof of its Proposition 4.10. The original version of Assumption 3.1in (Sablayrolles et al., 2019), quoted by Dong et al. (2022), is given below.

**Assumption 3.2 (Original Assumption in (Sablayrolles et al., 2019))** *Given the training dataset $\mathcal{S}$ and loss function $\ell$, the distribution of the model parameters is*

$$\mathbb{P}(\boldsymbol{\theta}|\mathcal{S}) \propto \exp(-\frac{1}{T}\sum_{i=1}^{|\mathcal{S}|}\ell(\boldsymbol{f_\theta}(\boldsymbol{s}_i),y_i)) \tag{6}$$

Comparing the above two assumptions, we observe that Assumption 3.1 has a controversial issue— Given the same loss function and model architecture, the posterior model parameter distribution assumed in Assumption 3.1 is fixed, regardless of the stochasticity and the learning method. Note that in an extreme case, if we use gradient ascent with fixed seeds, the parameter distribution could collapse into a Dirac distribution. In contrast, Eq. 6 in Assumption 3.2 has a temperature hyperparameter $T$ depending on the stochasticity and the learning method, *i.e.,* $T \to 0$ corresponds to MAP, and a small $T$ corresponds to averaged SGD (Polyak & Juditsky, 1992; Sablayrolles et al., 2019). From this viewpoint, although it is intractable to verify the correctness of Assumption 3.2, it is intuitive that Assumption 3.1 is controversial. Here a natural question to ask is—Could we instead use Assumption 3.2 in the proof of Proposition 4.10 in (Dong et al., 2022)? The answer is No because when $T$ is small, we could not approximate $\exp(\frac{1}{T}\sum_{i=1}^{|\mathcal{S}|}(\ell(\boldsymbol{f_\theta}(\boldsymbol{s}_i),y_i) - \ell(\boldsymbol{f_\theta}(\boldsymbol{s}_i'),y_i))) - 1$ by $\frac{1}{T}\sum_{i=1}^{|\mathcal{S}|}(\ell(\boldsymbol{f_\theta}(\boldsymbol{s}_i),y_i) - \ell(\boldsymbol{f_\theta}(\boldsymbol{s}_i'),y_i))$. Thus, we could not have Eq. 39 in the proof of Proposition 4.10 in (Dong et al., 2022) with Assumption 3.2.

In the following, we introduce the other controversial assumption which is *implicitly* used in the proof of Proposition 4.10 in (Dong et al., 2022) (Eq. 35 in (Dong et al., 2022)).

**Assumption 3.3 (Implicit Assumption in (Dong et al., 2022))** *With random initialization $\tilde{\boldsymbol{s}}_i \sim \mathcal{N}(\boldsymbol{0}, \boldsymbol{I}_d)$, the synthetic data $\tilde{\boldsymbol{s}}_i^*$ that minimizes the distribution matching loss (Eq. 3) is*

$$\tilde{\boldsymbol{s}}_i^* = \tilde{\boldsymbol{s}}_i + \frac{1}{|\mathcal{T}|}\sum_{j=1}^{|\mathcal{T}|}\tilde{\boldsymbol{x}}_j \ \ (under \ \mathcal{E}), \tag{7}$$

*where the data is represented under an orthogonal basis $\mathcal{E} = \{\boldsymbol{e}_1, \boldsymbol{e}_2, ..., \boldsymbol{e}_d\}$ with $\mathcal{E}_\mathcal{T} = \{\boldsymbol{e}_1, \boldsymbol{e}_2, ..., \boldsymbol{e}_{dim(span(\mathcal{T}))}\}$ forming the orthogonal basis of $span(\mathcal{T})$ (Dong et al., 2022).*

Let $\boldsymbol{Q}$ be the transformation matrix from $\mathcal{E}$ to the standard basis. We then have

$$\boldsymbol{s}_i^* = \boldsymbol{Q}\tilde{\boldsymbol{s}}_i + \frac{1}{|\mathcal{T}|}\sum_{j=1}^{|\mathcal{T}|}\boldsymbol{x}_j \ \ (under \ standard \ basis), \tag{8}$$

where $\boldsymbol{s}_i^*$ and $\boldsymbol{x}_j$ represent $\tilde{\boldsymbol{s}}_i^*$ and $\tilde{\boldsymbol{x}}_j$ under the standard basis. In Appendix C, we detail how to compute the transformation matrix and the synthetic data given in Eq. 8 for each class. Actually, we need to use $\mathcal{T}_c{}^*$ instead of $\mathcal{T}$ to compute the synthetic data for class $c$.

Assumption 3.3 is only proved on linear feature extractors, but it is directly used in the proof of Proposition 4.10 in (Dong et al., 2022) (Eq. 35 in (Dong et al., 2022)), which attempts to prove a general privacy bound for dataset condensation. Unfortunately, Assumption 3.3 is not always true when we use nonlinear feature extractors to learn the synthetic data.

We conduct an experiment to verify the above claim: We use the same data initialization to compute a set of 500 synthetic data samples with Eq. 8 (See Appendix C) and learn a set of 500 synthetic data samples by DM with nonlinear extractors on CIFAR10. We then compute the DM loss on the synthetic data by Eq. 3 with nonlinear extractors. The DM loss on the synthetic data computed with Eq. 8 is much higher than the DM loss on the DM-learned synthetic data, which conflicts with Assumption 3.3. We further train two ConvNet models on the two sets of synthetic data, respectively. The model trained on the synthetic data computed by Eq. 8 only has about $25\%$ accuracy[†], while the model trained on the DM-generated synthetic data achieves approximately $60\%$ accuracy.

Moreover, we test whether the distributions of the synthetic data computed by Eq. 8 and the synthetic data generated by DM with nonlinear extractors may have the same population mean for each class

---

[*]$\mathcal{T}_c$ is the subset of $\mathcal{T}$ that contains all the data with label $c$.

[†]Similar to the accuracy achieved by DM with linear extractors reported in (Dong et al., 2022)' appendix.

$c$. Specifically, we define the null hypothesis as

$$H_0 : \mu_c(\text{synthetic data defined by Eq. 8}) = \mu_c(\text{synthetic data generated by nonlinear DM})$$

We use BS test (Bai & Saranadasa, 1996) since the conventional Hotelling's $T^2$ test (Hotelling, 1992) is only applicable to the case that $d < N_1 + N_2 - 2$, where $d$ is the data dimension, and $N_1$ and $N_2$ refer to the number of samples from the two populations. We compute the test statistic of BS test, denoted by $Z_{BS}$ for the 10 classes of CIFAR10. The smallest $Z_{BS}$ is approximately $-3.03 < -z_{0.005}$[‡], which gives us sufficient evidence to reject the null hypothesis. Thus, even with the same random initialization, the synthetic data defined by Eq. 8 and the synthetic data generated by DM with nonlinear feature extractors can have completely different distributions. Therefore, it is problematic to use Assumption 3.3 to prove a general privacy bound for dataset condensation.

### 3.2 DIFFERENTIALLY PRIVATE DATASET CONDENSATION (DPDC)

We show that the privacy bound proved by Dong et al. (2022) is problematic due to two controversial assumptions. To correctly connect DC and DP, we propose two differentially private dataset condensation (DPDC) algorithms—a linear algorithm (LDPDC) and a nonlinear algorithm (NDPDC).

**Linear DPDC (LDPDC)**   We illustrate LDPDC in Algorithm 1. For each class $c$, we construct $M$ synthetic data samples $\{s_j^c\}_{j=1}^M$. For each synthetic sample $s_j^c$, we randomly sample a subset $\{x_k^c\}_{k=1}^{L_j^c}$ from $\mathcal{T}_c$ by Poisson Sampling with probability $L/N_c$. $L$ is the group size (Abadi et al., 2016), and $N_c = |\mathcal{T}_c|$. $L_j^c$ follows a Poisson distribution with expectation $L$. Similar to the $q = L/N$ in (Abadi et al., 2016) and the Opacus library, the sampling probability $q_c = L/N_c$ is fixed for each class in the execution of the algorithms—For the adjacent datasets of $\mathcal{T}_c$, we still consider $q_c$ as the sampling probability, then we could exploit the prior theoretical results on subsampling for DP analysis (similar to Opacus). With $\{x_k^c\}_{k=1}^{L_j^c}$, we compute $s_j^c$ by $s_j^c = \frac{1}{L}(\mathcal{N}(\mathbf{0}, \sigma^2 \mathbf{I}_d) + \sum_{k=1}^{L_j^c} x_k^c)$ where $\mathcal{N}(\mathbf{0}, \sigma^2 \mathbf{I}_d)$ refers to Gaussian random noise with standard deviation $\sigma$. We do not use a formula similar to Eq. 8 to construct synthetic data because $\mathbf{Q}$ leaks private information.

---

**Algorithm 1** Linear Differentially Private Dataset Condensation (LDPDC)

---

**Require:** Original Dataset $\mathcal{T} = \mathcal{T}_1 \cup \mathcal{T}_2 ... \cup \mathcal{T}_C$; the number of classes $C$; number of data samples per class $N_c$; number of synthetic samples per class $M$; group size $L$.

  **for** each class $c$ **do**

    **for** $j = 1$ to $M$ **do**

      Take a randomly sampled subset $D_c = \{x_k^c, c\}_{k=1}^{L_j^c}$ from $\mathcal{T}_c$ with sampling probability $L/N_c$ (by Poisson Sampling, similar to (Abadi et al., 2016; Yousefpour et al., 2021)) .

      $s_j^c = \frac{1}{L}(\mathcal{N}(\mathbf{0}, \sigma^2 \mathbf{I}_d) + \sum_{k=1}^{L_j^c} x_k^c)$

    **end for**

  **end for**

  Output the synthetic dataset $\mathcal{S} = \{\{s_j^c\}_{j=1}^M\}_{c=1}^C$

---

**Nonlinear DPDC (NDPDC)**   We illustrate NDPDC in Algorithm 2, which is designed upon the idea of matching the representations of original and synthetic data. We follow (Zhao & Bilen, 2021a) to use differentiable augmentation function $\mathcal{A}_{w_c}(\cdot)$[§] to boost the performance ($\Phi_\theta(\mathcal{A}_{w_c}(\cdot))$ is similar to a composite function). In each iteration of Algorithm 2, we first sample random parameters $\theta$ for the feature extractor $\Phi_\theta$ (not pretrained) and initialize the loss as 0. After that, for each class $c$, we sample the augmentation parameters $w_c$ and randomly sample a subset from $\mathcal{T}_c$ by Poisson sampling with sampling probability $L/N_c$. We then compute the representations of the original data and synthetic data and clip the representations with a pre-defined threshold $G$.

We remark that it is essential to clip both the representations of original and synthetic data: We clip the representations of the original data for the purpose of bounding the $\ell_2$ sensitivity; We also clip the representations of the synthetic data in order to match the representations on a similar scale. Since $G$ is pre-defined as the constant 1 (not computed from original data), the operation of clipping

---

[‡]$z_{0.005}$ refers to a critical value with 99% confidence level (two-tail).

[§]We refer the interested readers to (Zhao & Bilen, 2021b) for more details on $\mathcal{A}_{w_c}(\cdot)$.

the synthetic data representations (*i.e.*, $\tilde{\boldsymbol{r}}(\boldsymbol{s}_j^c) = \min(1, \frac{G}{\|\boldsymbol{r}(\boldsymbol{s}_j^c)\|_2})\boldsymbol{r}(\boldsymbol{s}_j^c)$ in Algorithm 2) does not leak private information regarding the original data. After clipping the representations, we add Gaussian noise to the sum of the clipped original data representations.

We use the squared $\ell_2$ distance between the randomized sum of the clipped original data representations (*i.e.*, $\mathcal{N}(\boldsymbol{0}, \sigma^2\boldsymbol{I}) + \sum_{i=1}^{|D_c|} \tilde{\boldsymbol{r}}(\boldsymbol{x}_i^c)$) and the sum of the clipped synthetic data representations multiplied by a factor $L/M$ (*i.e.*, $\frac{L}{M}\sum_{j=1}^{M}\tilde{\boldsymbol{r}}(\boldsymbol{s}_j^c)$) as the loss. We use the factor $L/M$ because $\sum_{i=1}^{|D_c|}\tilde{\boldsymbol{r}}(\boldsymbol{x}_i^c)$ sums up $|D_c|$ ($\mathbb{E}(|D_c|) = L$) representations, while $\sum_{j=1}^{M}\tilde{\boldsymbol{r}}(\boldsymbol{s}_j^c)$ sums up $M$ representations. At the end of each iteration, we update the synthetic data $\mathcal{S}$ with the gradient of the loss $\ell$ with respect to $\mathcal{S}$, similar to Algorithm 1 in (Zhao & Bilen, 2021a). *In practical implementation*, following (Zhao et al., 2021; Zhao & Bilen, 2021b;a), we implement $\mathcal{S}$ as a tensor variable with size $[N, \text{data\_shape}]$ (e.g., $[N, 3, 32, 32]$ on CIFAR10), where $N$ is the size of the synthetic dataset.

Here a natural question to ask is—Why not combine distribution matching and DP-SGD for differentially private data condensation? For many common deep learning tasks, we could compute sample-wise loss so that DP-SGD can clip the sample-wise loss gradients to bound the sensitivity. However, distribution matching uses the squared $\ell_2$ distance between the averaged original data representations and the averaged synthetic data representations as loss, so we could not directly compute sample-wise loss and apply DP-SGD to distribution matching.

---

**Algorithm 2** Nonlinear Differentially Private Dataset Condensation (NDPDC)

---

**Require:** Original Dataset $\mathcal{T} = \mathcal{T}_1 \cup \mathcal{T}_2 ... \cup \mathcal{T}_C$; the number of classes $C$; the number of data samples per class $N_c$; the number of synthetic samples per class $M$; feature extractors $\boldsymbol{\Phi}_{\boldsymbol{\theta}}$ (not pretrained); parameter distribution $P_{\boldsymbol{\theta}}$; group size $L$; number of iterations $I$.

  Initialize $\mathcal{S} = \{\{\boldsymbol{s}_j^c\}_{j=1}^{M}\}_{c=1}^{C}$ with random noise from $\mathcal{N}(\boldsymbol{0}, \boldsymbol{I}_d)$

  **for** each iteration (total number of iterations is $I$) **do**

    Randomly sample $\boldsymbol{\theta}$ from $P_{\boldsymbol{\theta}}$ and initialize the loss as $\ell = 0$

    **for** each class $c$ **do**

      Sample the augmentation parameters $\boldsymbol{w}_c$.

      Take a randomly sampled subset $D_c$ from $\mathcal{T}_c$ with sampling probability $L/N_c$ (by Poisson Sampling, similar to (Abadi et al., 2016; Yousefpour et al., 2021)).

      Compute Representations: $\boldsymbol{r}(\boldsymbol{x}_i^c) = \boldsymbol{\Phi}_{\boldsymbol{\theta}}(\mathcal{A}_{\boldsymbol{w}_c}(\boldsymbol{x}_i^c))$ for the subset $D_c = \{\boldsymbol{x}_i^c, c\}_{i=1}^{|D_c|}$; $\boldsymbol{r}(\boldsymbol{s}_j^c) = \boldsymbol{\Phi}_{\boldsymbol{\theta}}(\mathcal{A}_{\boldsymbol{w}_c}(\boldsymbol{s}_j^c))$ for $S_c = \{\boldsymbol{s}_j^c\}_{j=1}^{M}$.

      Norm Clipping: $\tilde{\boldsymbol{r}}(\boldsymbol{s}_j^c) = \min(1, \frac{G}{\|\boldsymbol{r}(\boldsymbol{s}_j^c)\|_2})\boldsymbol{r}(\boldsymbol{s}_j^c)$; $\tilde{\boldsymbol{r}}(\boldsymbol{x}_i^c) = \min(1, \frac{G}{\|\boldsymbol{r}(\boldsymbol{x}_i^c)\|_2})\boldsymbol{r}(\boldsymbol{x}_i^c)$.

      Compute Loss: $\ell = \ell + \|\frac{L}{M}\sum_{j=1}^{M}\tilde{\boldsymbol{r}}(\boldsymbol{s}_j^c) - (\mathcal{N}(\boldsymbol{0}, \sigma^2\boldsymbol{I}) + \sum_{i=1}^{|D_c|}\tilde{\boldsymbol{r}}(\boldsymbol{x}_i^c))\|_2^2$.

    **end for**

    $\mathcal{S} = \mathcal{S} - \eta\nabla_{\mathcal{S}}\ell$ ($\boldsymbol{s}_j^c = \boldsymbol{s}_j^c - \eta\nabla_{\boldsymbol{s}_j^c}\ell \;\forall \boldsymbol{s}_j^c \in \mathcal{S}$).

  **end for**

  Output the synthetic dataset $\mathcal{S} = \{\{\boldsymbol{s}_j^c\}_{j=1}^{M}\}_{c=1}^{C}$

---

Theorem 3.1 and Theorem 3.2 bound the privacy risk of Algorithm 1 and Algorithm 2, respectively.

**Theorem 3.1** *Suppose the original data $\boldsymbol{x}$ satisfies $\boldsymbol{x} \in [-b, b]^d$, and let $\Omega_{q, \tilde{\sigma}_1}(\alpha) \triangleq \mathcal{D}_{\alpha}((1 - q)\mathcal{N}(0, \tilde{\sigma}_1^2) + q\mathcal{N}(1, \tilde{\sigma}_1^2)\|\mathcal{N}(0, \tilde{\sigma}_1^2))$ with $\tilde{\sigma}_1 = \sigma/(b\sqrt{d})$ and $q = \max_c(L/N_c)$, then LDPDC obeys $(\alpha, M\Omega_{q, \tilde{\sigma}_1}(\alpha))$-RDP and $(\min_{\alpha>1}(M\Omega_{q, \tilde{\sigma}_1}(\alpha) + \log((\alpha-1)/\alpha) - (\log \delta + \log \alpha)/(\alpha-1)), \delta)$-DP.*

**Theorem 3.2** *Let $\Omega_{q, \tilde{\sigma}_2}(\alpha) \triangleq \mathcal{D}_{\alpha}((1 - q)\mathcal{N}(0, \tilde{\sigma}_2^2) + q\mathcal{N}(1, \tilde{\sigma}_2^2)\|\mathcal{N}(0, \tilde{\sigma}_2^2))$ with $\tilde{\sigma}_2 = \sigma/G$ and $q = \max_c(L/N_c)$, then NDPDC obeys $(\alpha, I\Omega_{q, \tilde{\sigma}_2}(\alpha))$-RDP and $(\min_{\alpha>1}(I\Omega_{q, \tilde{\sigma}_2}(\alpha) + \log((\alpha-1)/\alpha) - (\log \delta + \log \alpha)/(\alpha-1)), \delta)$-DP, where $I$ is the number of iterations.*

We detail the proof of Theorem 3.1 and Theorem 3.2 in Appendix A. The basic idea is to prove the RDP bound on $\mathcal{T}_c = \{\boldsymbol{x}_i, c\}_{i=1}^{N_c}$, where we only need to consider $\boldsymbol{x}_i$ due to the same $c$, followed by generalizing the theoretical result to $\mathcal{T}$ using Lemma 2.3. In the experiments, we follow Opacus to exploit Mironov et al. (2019)'s method for computing $\Omega_{q, \sigma}(\alpha)$. We note that Mironov et al. (2019) did not provide convergence analysis for their computation method when $\alpha$ is a fractional number. For completeness, we present our convergence analysis for the computation method in Appendix B.

| Dataset → | MNIST | | FMNIST | |
|---|---|---|---|---|
| Method ↓ | Test Acc | DP Budget | Test Acc | DP Budget |
| DM with Rand Init | $98.38\% \pm 0.05\%$ | No | $86.90\% \pm 0.44\%$ | No |
| DP Sinkhorn | $86.92\% \pm 0.93\%$ | $(10, 10^{-5})$-DP | $65.60\% \pm 1.06\%$ | $(10, 10^{-5})$-DP |
| DP-MERF | $84.81\% \pm 2.04\%$ | $(1, 10^{-5})$-DP | $63.05\% \pm 2.05\%$ | $(1, 10^{-5})$-DP |
| Linear DPDC (LDPDC) | $85.79\% \pm 0.81\%$ | $(1.10, 10^{-5})$-DP | $63.95\% \pm 0.42\%$ | $(1.06, 10^{-5})$-DP |
| Nonlinear DPDC (NDPDC) | $97.35\% \pm 0.13\%$ | $(6.12, 10^{-5})$-DP | $82.72\% \pm 0.35\%$ | $(5.45, 10^{-5})$-DP |
| Dataset → | CIFAR10 | | CelebA | |
| Method ↓ | Test Acc | DP Budget | Test Acc | DP Budget |
| DM with Rand Init | $59.69\% \pm 0.44\%$ | No | $84.13\% \pm 0.42\%$ | No |
| DP Sinkhorn | $15.09\% \pm 0.33\%$ | $(10, 10^{-5})$-DP | $71.72\% \pm 1.13\%$ | $(10, 10^{-5})$-DP |
| DP-MERF | $17.10\% \pm 0.78\%$ | $(1, 10^{-5})$-DP | $69.26\% \pm 0.90\%$ | $(1, 10^{-5})$-DP |
| Linear DPDC (LDPDC) | $25.81\% \pm 0.52\%$ | $(1.14, 10^{-5})$-DP | $68.72\% \pm 2.26\%$ | $(0.61, 10^{-5})$-DP |
| Nonlinear DPDC (NDPDC) | $52.68\% \pm 0.40\%$ | $(6.72, 10^{-5})$-DP | $80.66\% \pm 0.63\%$ | $(0.71, 10^{-5})$-DP |

Table 1: We use the default settings for all the methods. We employ 50 synthetic samples per class to train ConvNet models and report the testing accuracy here. We follow (Zhao & Bilen, 2021a) to apply the augmentations in (Zhao & Bilen, 2021b) when training ConvNet models. According to Table 8 in Appendix D, even using low DP budgets ($\epsilon < 1$), NDPDC still outperforms LDPDC, DP-MERF, and DP-Sinkhorn. Similar to DP-Sinkhorn, we can also fix a target DP budget and compute the corresponding $\sigma$ (or $I$) to run LDPDC and NDPDC.

| | MNIST | FMNIST | CIFAR10 | CelebA |
|---|---|---|---|---|
| Linear DPDC (LDPDC) | $85.80\% \pm 0.39\%$ | $63.64\% \pm 0.76\%$ | $25.46\% \pm 0.83\%$ | $69.64\% \pm 1.01\%$ |
| Nonlinear DPDC (NDPDC) | $95.32\% \pm 0.29\%$ | $78.79\% \pm 0.37\%$ | $42.40\% \pm 0.86\%$ | $81.47\% \pm 0.80\%$ |
| DP Sinkhorn | $55.43\% \pm 1.54\%$ | $43.22\% \pm 1.40\%$ | $12.62\% \pm 0.27\%$ | $64.02\% \pm 0.48\%$ |
| DP-MERF | $84.81\% \pm 2.04\%$ | $63.05\% \pm 2.05\%$ | $17.10\% \pm 0.78\%$ | $69.26\% \pm 0.90\%$ |

Table 2: We set $\epsilon = 1$ and compare LDPDC, NDPDC, DP Sinkhorn, DP-MERF.

## 4 EXPERIMENTS

### 4.1 EXPERIMENTAL SETUP

We follow (Zhao & Bilen, 2021b;a; Dong et al., 2022) to conduct experiments on four widely-used datasets: MNIST (Deng, 2012), Fashion-MNIST (Xiao et al., 2017), CIFAR10 (Krizhevsky et al., 2009), and CelebA (gender classification) (Liu et al., 2015). In the following, we introduce the baselines, DPDC settings, and the method for evaluating the synthetic data utility. We provide DPDC's code in the supplementary material, where the readers could find more technical details.

**Baselines** We compare DPDC with the state-of-the-art dataset condensation method, *i.e.,* distribution matching (DM) (Zhao & Bilen, 2021a), and two recent DP generative methods, *i.e.,* DP-Sinkhorn (Cao et al., 2021) and DP-MERF (Harder et al., 2021). For DP-Sinkhorn, we use Cao et al. (2021)'s code[¶] to run the experiments. We set $m$ to 1 and the target $\epsilon$ to 10. For DP-MERF, we use Harder et al. (2021)'s code[‖] and follow (Harder et al., 2021) to set $\sigma$ to 5.

**DPDC Settings** For LDPDC, we set $\sigma = \sqrt{d}$, $M = 50$, $L = 50$ by default. Given that $b = 1$, $\tilde{\sigma}_1 = \sigma/(b\sqrt{d}) = 1$. For NDPDC, the default settings are $\sigma = 1$, $G = 1$, $M = 50$, $L = 50$, $I = 10000$, and $\eta = 1$. We set $G = 1$, so $\tilde{\sigma}_2 = \sigma/G = 1$. We follow (Zhao & Bilen, 2021a) to use three-layer convolutional neural networks as the feature extractors (also called ConvNet-based feature extractors) for NDPDC. Batch normalization (BN) (Ioffe & Szegedy, 2015) is not DP friendly since a sample's normalized value depends on other samples (Yousefpour et al., 2021). Therefore, we do not use BN in the extractors. Since the data statistics like channel-wise means and channel-wise standard deviation may leak private information, we follow (Cao et al., 2021) to use a fixed value 0.5 for normalizing the images, which does not make obvious difference in the performance of DPDC and the baselines. After normalization, the pixel values range from $-1$ to 1.

**Performance Evaluation** We employ the evaluation method in (Zhao & Bilen, 2021b;a; Dong et al., 2022) to compare the performance of DM, DP-Sinkhorn, DP-MERF, and our DPDC algo-

---

[¶] https://github.com/nv-tlabs/DP-Sinkhorn_code
[‖] https://github.com/ParkLabML/DP-MERF/tree/master/code_balanced

| LDPDC | MLP | LeNet | AlexNet | VGG11 | ResNet18 |
|--------|-----|-------|---------|-------|----------|
| MNIST | $80.40\% \pm 0.33\%$ | $83.32\% \pm 3.03\%$ | $83.22\% \pm 0.67\%$ | $85.81\% \pm 0.74\%$ | $88.14\% \pm 0.73\%$ |
| FMNIST | $68.84\% \pm 0.58\%$ | $68.49\% \pm 1.36\%$ | $66.45\% \pm 0.94\%$ | $67.31\% \pm 1.01\%$ | $67.58\% \pm 0.92\%$ |
| CIFAR10 | $30.60\% \pm 0.18\%$ | $29.82\% \pm 0.81\%$ | $29.65\% \pm 0.82\%$ | $24.86\% \pm 0.29\%$ | $23.79\% \pm 0.59\%$ |
| CelebA | $69.48\% \pm 1.51\%$ | $66.95\% \pm 0.53\%$ | $67.61\% \pm 0.87\%$ | $66.87\% \pm 2.23\%$ | $65.88\% \pm 1.42\%$ |

| NDPDC | MLP | LeNet | AlexNet | VGG11 | ResNet18 |
|--------|-----|-------|---------|-------|----------|
| MNIST | $93.15\% \pm 0.83\%$ | $96.82\% \pm 0.13\%$ | $97.00\% \pm 0.32\%$ | $97.37 \pm 0.13\%$ | $97.67\% \pm 0.14\%$ |
| FMNIST | $79.98\% \pm 0.33\%$ | $80.89\% \pm 0.40\%$ | $81.94\% \pm 0.54\%$ | $82.63\% \pm 0.54\%$ | $81.50\% \pm 0.33\%$ |
| CIFAR10 | $36.99\% \pm 0.62\%$ | $36.94\% \pm 1.24\%$ | $42.60\% \pm 1.06\%$ | $48.80\% \pm 0.24\%$ | $44.79\% \pm 0.64\%$ |
| CelebA | $75.95\% \pm 1.30\%$ | $78.02\% \pm 1.01\%$ | $79.69\% \pm 0.91\%$ | $81.25\% \pm 1.31\%$ | $82.32\% \pm 0.68\%$ |

Table 3: Performance on varied model architectures with default settings: For NDPDC, the synthetic data is learned on ConvNet-based feature extractors and evaluated on those model architectures. The privacy budgets and the results on ConvNet are given in Table 1.

rithms. The evaluation method is to train deep learning models on the synthetic data from scratch and compare their accuracy on the real testing data. Higher testing accuracy indicates better synthetic data utility for training deep learning models (better performance). We also follow (Zhao & Bilen, 2021b;a) to train a variety of model architectures, including MLP (Haykin, 1994), LeNet (LeCun et al., 1998), AlexNet (Krizhevsky et al., 2017), VGG11 (Simonyan & Zisserman, 2014), and ResNet18 (He et al., 2016), on DPDC-generated synthetic data to evaluate the data utility.

## 4.2 MAIN RESULTS

We report the main experimental results in Table 1: Our LDPDC algorithm achieves comparable performance to DP-MERF and DP-Sinkhorn with low DP budgets. Our NDPDC algorithm provides acceptable DP guarantees ($\epsilon < 10$) with a mild utility loss, compared to the random-initialized non-private DM method (Zhao & Bilen, 2021a). Furthermore, NDPDC allows a flexible trade-off between synthetic data utility and DP budget—If we are not satisfied with NDPDC's DP budgets in Table 1, we could increase $\sigma$ to reduce the budget. As shown in Table 2, even with low DP budgets $\epsilon = 1$, NDPDC still outperforms LDPDC, DP-Sinkhorn, and DP-MERF. For DP-MERF, even if we decrease $\sigma$ to $0.5$ ($\epsilon > 10$), the accuracy increment is only about $7\%$ on FMNIST and less than $5\%$ on the other datasets, as shown in Table 9 in Appendix D. We conjecture that a small amount of NDPDC-condensed synthetic data is more useful than a small amount of DP-generator generated synthetic data because DP generative methods optimize the generative model parameters, while NDPDC directly optimizes the small amount of synthetic data. It is worth noting that DPMix (Lee et al., 2019) is a recent linear DP data publishing method, which seems similar to LDPDC. However, LDPDC and DPMix still have some differences as discussed in Appendix F, indicating that LDPDC may be better than DPMix. Compared to LDPDC and DPMix, NDPDC is more suitable for solving nonlinear problems such as image recognition. According to the results in (Lee et al., 2019), NDPDC outperforms DPMix by 13%-24% in testing accuracy on MNIST and CIFAR10.

We further train a variety of model architectures on the synthetic data generated by LDPDC and NDPDC and report the testing accuracy in Table 3. Since LDPDC does not rely on deep networks to learn the synthetic data, it is hard to predict which network architecture can make the best use of the LDPDC-generated synthetic data. According to the results in Table 3, on FMNIST, CIFAR10, and CelebA, MLP makes the best use of the simple LDPDC-generated synthetic data, while on MNIST, ResNet18 makes the best use of the synthetic data. For NDPDC, since the synthetic data is learned on ConvNet-based extractors, ConvNet makes better use of the synthetic data than the other architectures. Besides, we visualize the synthetic images in Appendix D: NDPDC-generated images look more diverse and more useful than the synthetic images generated by DP-Sinkhorn, DP-MERF, and DPMix (Fig. 1 in (Lee et al., 2019) provides synthetic images generated by DPMix.).

## 4.3 ABLATION STUDY ON NDPDC

Although LDPDC is simple and comparable to recent DP generative methods here, we still recommend the readers to use NDPDC in practice because of its outstanding performance. In this subsection, we conduct ablation study for NDPDC on MNIST, FMNIST, and CIFAR10 with recommendations on how to select hyperparameters for executing NDPDC. When we study the effects of one hyperparameter, we fix the other hyperparameters as the default settings.

**Effects of Noise Multiplier $\sigma$ on Privacy and Utility** We plot the DP budgets of NDPDC with different $\sigma$ in Fig. 1 in Appendix D and report the corresponding testing accuracy in Table 4. Fig. 1 in Appendix D and Table 4 indicate that, as $\sigma$ increases, $\epsilon$ will decrease, and the testing accuracy will also decrease. But the testing accuracy does not decrease much as $\sigma$ increases. Thus, if we are unsatisfied with the DP budget, we could simply increase $\sigma$ to obtain a low DP budget with a little

| Noise | MNIST | FMNIST | CIFAR10 |
|---|---|---|---|
| $\sigma = 0.75$ | 97.51% | 83.28% | 54.33% |
| $\sigma = 1.25$ | 97.06% | 82.23% | 51.36% |
| $\sigma = 1.5$ | 96.86% | 81.66% | 50.12% |
| $\sigma = 2.0$ | 96.46% | 80.96% | 47.93% |

Table 4: The averaged testing accuracy of ConvNets trained on the synthetic data generated by NDPDC with different noise multiplier $\sigma$.

loss of synthetic data utility. Additionally, we do not recommend to set $\sigma/G \leq 0.75$, otherwise $\epsilon$ will be larger than 10, then the DP guarantee is not very useful.

**Effects of Group Size $L$ on Privacy and Utility** We plot the DP budgets with different $L$ in Fig. 1 in Appendix D and show the testing accuracy in Table 5. If we increase $L$, it is expected that more original data will be sampled in each step for learning the synthetic data, and thus, both $\epsilon$ and the testing accuracy will increase. According to Fig. 1 in Appendix D and Table 5, if we are unsatisfied with the DP budgets, we could also

| Group Size | MNIST | FMNIST | CIFAR10 |
|---|---|---|---|
| $L = 25$ | 96.41% | 80.88% | 48.07% |
| $L = 75$ | 97.61% | 83.97% | 54.81% |
| $L = 100$ | 97.90% | 84.40% | 56.42% |

Table 5: The averaged testing accuracy of ConvNets trained on the synthetic data generated by NDPDC with different group size $L$.

decrease $L$ to 25 to obtain better DP budgets with a minor utility loss.

**Effects of Number of Iterations $I$ on Privacy and Utility** We plot the DP budgets in Fig. 1 in Appendix D and show the testing accuracy in Table 6. If we increase $I$, since NDPDC will learn on the original data for more iterations, both the DP budget and testing accuracy will increase. Since the testing accuracy does not increase much when $I$ is large, reducing $I$ is another good choice beyond increasing $\sigma$ and reducing $L$ to save DP budget with acceptable utility loss.

| Iterations | MNIST | FMNIST | CIFAR10 |
|---|---|---|---|
| $I = 2000$ | 96.32% | 80.45% | 47.42% |
| $I = 4000$ | 96.84% | 81.64% | 50.07% |
| $I = 6000$ | 97.14% | 82.18% | 51.20% |
| $I = 8000$ | 97.28% | 82.50% | 52.19% |

Table 6: The averaged testing accuracy of ConvNets trained on the synthetic data generated by NDPDC with different number of iterations $I$.

**Effects of Synthetic Data Size $M$ on Privacy and Utility** We show the testing accuracy with different $M$ in Table 7: As $M$ increases from 10 to 200, the testing accuracy also increases but will almost stop the uptrend around a certain $M$. This is probably because more synthetic data has the potential to capture more information, but the DP guarantee also limits the information leaked from the original data. Since the DP budget is

| Data Size | MNIST | FMNIST | CIFAR10 |
|---|---|---|---|
| $M = 10$ | 96.47% | 80.00% | 45.88% |
| $M = 100$ | 97.54% | 82.88% | 54.24% |
| $M = 200$ | 97.50% | 83.16% | 54.26% |

Table 7: The averaged testing accuracy of ConvNets achieved by NDPDC with different number of synthetic samples per class $M$.

unchanged, after the amount of synthetic data is enough for capturing all the limited information, more synthetic data could not capture more useful information. Given the experimental results, $M = 50$ is a good setting here. We do not recommend setting a larger $M$ ($M > 50$), which will result in marginal performance gain but much more cost in the downstream applications.

## 5 CONCLUSION

In this paper, we revisit a recent proposal in ICML'22 on privacy-preserving dataset condensation and reveal that its proposed privacy bound is problematic because of two controversial assumptions. To correctly connect data condensation and differential privacy, we propose two differentially private dataset condensation (DPDC) algorithms—LDPDC and NDPDC. We demonstrate that LDPDC can use low DP budgets to achieve comparable performance to DP-Sinkhorn and DP-MERF. Moreover, we show that NDPDC can provide DP guarantees with a mild utility loss compared to the distribution matching method. We hope our work can inspire further research in this potential direction to alleviate the cost burden and privacy concern in deep learning.

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

## A  OMITTED PROOF

We first prove Lemma A.1. Based on Lemma A.1, we can easily prove Corollary A.1, which will be used in the proof of Theorem 3.1 & 3.2.

**Lemma A.1** *Let $u(z)$ and $\nu(z)$ be two differentiable probability density functions on a domain $\mathcal{Z}$ $(u, \nu : \mathcal{Z} \mapsto \mathbb{R})$. If $u(z) \neq \nu(z)$ and $u(z), \nu(z) > 0$ on $\mathcal{Z}$, then $\mathcal{D}_\alpha((1 - q)u(z) + q\nu(z) \| u(z))$ is an increasing function w.r.t. $q$ when $\alpha > 1$ and $q \in [0, 1]$.*

Lemma A.1 is easy to understand: As $q$ increases, the weight of $u(z)$ in the mixture $(1 - q)u(z) + q\nu(z)$ decreases, thus the divergence between $(1 - q)u(z) + q\nu(z)$ and $u(z)$ should increase. To our best knowledge, we are the first to present Lemma A.1, so we detail the proof in the following.

**Proof** [proof of Lemma A.1] The Rényi divergence $\mathcal{D}_\alpha((1 - q)u(z) + q\nu(z) \| u(z))$ is defined as

$$\frac{1}{\alpha - 1} \ln \int_{\mathcal{Z}} u(z) (\frac{(1 - q)u(z) + q\nu(z)}{u(z)})^\alpha dz = \frac{1}{\alpha - 1} \ln \int_{\mathcal{Z}} u(z)(1 - q + q\frac{\nu(z)}{u(z)})^\alpha dz \quad (9)$$

The derivative of Eq. 9 w.r.t $q$ is

$$\frac{1}{(\alpha - 1) \int_{\mathcal{Z}} u(z)(1 - q + q\frac{\nu(z)}{u(z)})^\alpha dz} \int_{\mathcal{Z}} \alpha(\nu(z) - u(z))(1 - q + q\frac{\nu(z)}{u(z)})^{\alpha - 1} dz, \quad (10)$$

To prove Lemma A.1, we need to show Eq. 10 is positive when $q \in [0, 1]$. Since $\alpha > 1$ and $u(z)(1 - q + q\frac{\nu(z)}{u(z)})^\alpha > 0$ (If $q \neq 0$, $1 - q + q\frac{\nu(z)}{u(z)} > 1 - q \geq 0$), we only need to prove $\int_{\mathcal{Z}} (\nu(z) - u(z))(1 - q + q\frac{\nu(z)}{u(z)})^{\alpha - 1} dz > 0$.

We divide $\mathcal{Z}$ into $\mathcal{Z}_1$ and $\mathcal{Z}_2$, where $\mathcal{Z}_1 = \{z \in \mathcal{Z} | \nu(z) < u(z)\}$ and $\mathcal{Z}_2 = \{z \in \mathcal{Z} | \nu(z) \geq u(z)\}$. Apparently, $\mathcal{Z}_1$ and $\mathcal{Z}_2$ are disjoint, and $\mathcal{Z} = \mathcal{Z}_1 \cup \mathcal{Z}_2$. Thus, we can rewrite $\int_{\mathcal{Z}} (\nu(z) - u(z))(1 - q + q\frac{\nu(z)}{u(z)})^{\alpha - 1} dz$ as

$$\int_{\mathcal{Z}_1} (\nu(z) - u(z))(1 - q + q\frac{\nu(z)}{u(z)})^{\alpha - 1} dz + \int_{\mathcal{Z}_2} (\nu(z) - u(z))(1 - q + q\frac{\nu(z)}{u(z)})^{\alpha - 1} dz \quad (11)$$

When $z \in \mathcal{Z}_1$, we have (i) $\nu(z) - u(z) < 0$; (ii) $\frac{\nu(z)}{u(z)} < 1$ $(0 < \nu(z) < u(z))$; (iii) $0 < (1 - q + q\frac{\nu(z)}{u(z)})^{\alpha - 1} < 1$ $(1 = 1 - q + q > 1 - q + q\frac{\nu(z)}{u(z)} > 1 - q \geq 0)$. Therefore,

$$\int_{\mathcal{Z}_1} (\nu(z) - u(z))(1 - q + q\frac{\nu(z)}{u(z)})^{\alpha - 1} dz > \int_{\mathcal{Z}_1} (\nu(z) - u(z)) dz \quad (12)$$

When $z \in \mathcal{Z}_2$, we have (i) $\nu(z) - u(z) \geq 0$; (ii) $\frac{\nu(z)}{u(z)} \geq 1$ $(0 < u(z) \leq \nu(z))$; (iii) $(1 - q + q\frac{\nu(z)}{u(z)})^{\alpha - 1} \geq 1$. Therefore,

$$\int_{\mathcal{Z}_2} (\nu(z) - u(z))(1 - q + q\frac{\nu(z)}{u(z)})^{\alpha - 1} dz \geq \int_{\mathcal{Z}_2} (\nu(z) - u(z)) dz \quad (13)$$

As a result,

$$\int_{\mathcal{Z}} (\nu(z) - u(z))(1 - q + q\frac{\nu(z)}{u(z)})^{\alpha - 1} dz > \int_{\mathcal{Z}_1} (\nu(z) - u(z)) dz + \int_{\mathcal{Z}_2} (\nu(z) - u(z)) dz$$

$$= \int_{\mathcal{Z}} (\nu(z) - u(z)) dz = \int_{\mathcal{Z}} \nu(z) dz - \int_{\mathcal{Z}} u(z) dz = 0$$

Thus, the derivative of Eq. 9 w.r.t $q$ is positive. This concludes the proof of Lemma A.1. ∎

**Corollary A.1** *Let $\Omega_{q_c, \sigma}(\alpha) \triangleq \mathcal{D}_\alpha((1 - q_c)\mathcal{N}(0, \sigma^2) + q_c\mathcal{N}(1, \sigma^2) \| \mathcal{N}(0, \sigma^2))$, where $c = 1, 2, ..., C$. We have $\max_c \Omega_{q_c, \sigma}(\alpha) = \Omega_{\max_c(q_c), \sigma}(\alpha)$.*

**Proof** [proof of Corollary A.1] Let $u(z) \triangleq \mathcal{N}(0, \sigma^2)$ and $\nu(z) \triangleq \mathcal{N}(1, \sigma^2)$ ($\mathcal{Z} \triangleq R$), then based on Lemma A.1, we know that $\Omega_{q,\sigma}(\alpha)$ is an increasing function w.r.t. $q$. Thus, the maximum of $\Omega_{q_c,\sigma}(\alpha)$ is achieved at $\max_c(q_c)$. This concludes the proof of Corollary A.1. ∎

In addition to Corollary A.1, we also need to use a definition and a lemma from (Mironov et al., 2019) in the proof of Theorem 3.1 & 3.2:

**Definition A.1 ((Mironov et al., 2019))** *Let $\boldsymbol{f}$ be a function mapping subsets of a dataset $D$ to $\mathbb{R}^n$. Mironov et al. (2019) define Sampled Gaussian Mechanism (SGM) as*

$$SGM_{q,\sigma}(D) = \boldsymbol{f}(\text{a subset sampled from } D \text{ with probability } q) + \mathcal{N}(\boldsymbol{0}, \sigma^2 \boldsymbol{I}^n) \tag{14}$$

**Lemma A.2 ((Mironov et al., 2019))** *Given the notations in Definition A.1, if for any two adjacent subsets $D_1$ and $D_2$ sampled from $D$, $\|\boldsymbol{f}(D_1) - \boldsymbol{f}(D_2)\| \leq 1$, then $SGM_{q,\sigma}(D)$ obeys $(\alpha, \epsilon)$-RDP, where $\epsilon = \mathcal{D}_\alpha((1-q)\mathcal{N}(0, \sigma^2) + q\mathcal{N}(1, \sigma^2) \| \mathcal{N}(0, \sigma^2))$.*

**Proof** [proof of Lemma A.2] Given the proof of Theorem 4 in (Mironov et al., 2019), we know that $SGM_{q,\sigma}(D)$ obeys $(\alpha, \epsilon)$-RDP, where $\epsilon$ could be $\max\{\mathcal{D}_\alpha((1-q)\mathcal{N}(0, \sigma^2) + q\mathcal{N}(1, \sigma^2) \| \mathcal{N}(0, \sigma^2)), \mathcal{D}_\alpha(\mathcal{N}(0, \sigma^2) \| (1-q)\mathcal{N}(0, \sigma^2) + q\mathcal{N}(1, \sigma^2))\}$.

According to Theorem 5 in (Mironov et al., 2019), $\mathcal{D}_\alpha((1-q)\mathcal{N}(0, \sigma^2) + q\mathcal{N}(1, \sigma^2) \| \mathcal{N}(0, \sigma^2)) \geq \mathcal{D}_\alpha(\mathcal{N}(0, \sigma^2) \| (1-q)\mathcal{N}(0, \sigma^2) + q\mathcal{N}(1, \sigma^2))$. Thus, $\epsilon = \Omega_{q,\sigma}(\alpha)$, where $\Omega_{q,\sigma}(\alpha) \triangleq \mathcal{D}_\alpha((1-q)\mathcal{N}(0, \sigma^2) + q\mathcal{N}(1, \sigma^2) \| \mathcal{N}(0, \sigma^2))$. ∎

Eventually, we can present the proof of our theorems.

**Proof** [proof of Theorem 3.1] Define $\boldsymbol{g}(D) = \sum_{i=1}^{|D|} \boldsymbol{x}_i$, where $D = \{\boldsymbol{x}_i, c\}_{i=1}^{|D|}$; $\boldsymbol{g}(\emptyset) = \boldsymbol{0}$. For any two adjacent datasets $D, D' \subset \mathcal{T}_c$ ($D' = D \cup \{\boldsymbol{x}'\}$), we have

$$\|\boldsymbol{g}(D) - \boldsymbol{g}(D')\|_2 = \|\boldsymbol{x}'\|_2 \leq b\sqrt{d}. \tag{15}$$

Thus, $\|\frac{1}{b\sqrt{d}}\boldsymbol{g}(D) - \frac{1}{b\sqrt{d}}\boldsymbol{g}(D')\|_2 \leq 1$.

In Algorithm 1, a synthetic data sample can be represented by

$$\boldsymbol{s}_i^c = \frac{b\sqrt{d}}{L}(\frac{1}{b\sqrt{d}}\boldsymbol{g}(D_c) + \mathcal{N}(\boldsymbol{0}, \tilde{\sigma}_1^2 \boldsymbol{I})), \text{ where } \tilde{\sigma}_1 = \sigma/(b\sqrt{d}) \tag{16}$$

where $D_c$ is a subset sampled from $\mathcal{T}_c$ with by Poisson sampling with sampling probability $q_c = L/N_c$. Apparently, $SG_{q,\tilde{\sigma}_1}(\mathcal{T}_c) = \frac{1}{b\sqrt{d}}\boldsymbol{g}(D_c) + \mathcal{N}(\boldsymbol{0}, \tilde{\sigma}_1^2 \boldsymbol{I})$ is a sampled Gaussian mechanism (SGM), and $\|\frac{1}{b\sqrt{d}}\boldsymbol{g}(D) - \frac{1}{b\sqrt{d}}\boldsymbol{g}(D')\|_2 \leq 1$.

Given Lemma A.2, $SG_{q,\tilde{\sigma}_1}(\mathcal{T}_c)$ obeys $(\alpha, \Omega_{q_c,\tilde{\sigma}_1}(\alpha))$-RDP with $\tilde{\sigma}_1 = \sigma/(b\sqrt{d})$ and $q_c = L/N_c$. According to the post-processing property, $\boldsymbol{s}_c$ also guarantees $(\alpha, \Omega_{q_c,\tilde{\sigma}_1}(\alpha))$-RDP.

Since Algorithm 1 outputs $M$ synthetic samples for each class $c$, according to Lemma 2.2, Algorithm 1 guarantees $(\alpha, M\Omega_{q_c,\tilde{\sigma}_1}(\alpha))$-RDP for $\mathcal{T}_c$. Since $\mathcal{T}_1, \mathcal{T}_2, ... \mathcal{T}_C$ are disjoint, according to Lemma 2.3, Algorithm 1 guarantees $(\alpha, \max_c(M\Omega_{q_c,\tilde{\sigma}_1}(\alpha)))$-RDP for $\mathcal{T} = \mathcal{T}_1 \cup \mathcal{T}_2 ... \cup \mathcal{T}_C$.

Corollary A.1 indicates that $\max_c(M\Omega_{q_c,\tilde{\sigma}_1}(\alpha)) = M\Omega_{\max_c(q_c),\tilde{\sigma}_1}(\alpha)$. Therefore, Algorithm 1 obeys $(\alpha, M\Omega_{q,\tilde{\sigma}_1}(\alpha))$-RDP with $q = \max_c(L/N_c)$ and $\tilde{\sigma}_1 = \sigma/(b\sqrt{d})$. ∎

**Proof** [Proof of Theorem 3.2] For $D = \{\boldsymbol{x}_i, y_i\}_{i=1}^{|D|}$, define $\boldsymbol{g}(D) = \sum_{i=1}^{|D|} \min(1, \frac{G}{\|\boldsymbol{r}(\boldsymbol{x}_i)\|_2})\boldsymbol{r}(\boldsymbol{x}_i)$, where $\boldsymbol{r}(\boldsymbol{x}_i) = \boldsymbol{\Phi}_{\boldsymbol{\theta}}(\mathcal{A}_{\boldsymbol{w}_c}(\boldsymbol{x}_i))$. Also, define $\boldsymbol{g}(\emptyset) = \boldsymbol{0}$. For any two adjacent subsets $D, D' \subset \mathcal{T}_c$, and $D' = D \cup \{\boldsymbol{x}'\}$, we have

$$\|\boldsymbol{g}(D) - \boldsymbol{g}(D')\|_2 = \|\boldsymbol{x}'\|_2 = \|\min(1, \frac{G}{\|\boldsymbol{r}(\boldsymbol{x}')\|_2})\boldsymbol{r}(\boldsymbol{x}')\|_2 \leq G. \tag{17}$$

Thus, $\|\frac{1}{G}\boldsymbol{g}(D) - \frac{1}{G}\boldsymbol{g}(D')\|_2 \leq 1$.

In each iteration of Algorithm 2, we rewrite $\mathcal{N}(\boldsymbol{0}, \sigma^2 \boldsymbol{I}) + \sum_{i=1}^{|D_c|} \tilde{\boldsymbol{r}}(\boldsymbol{x}_i^c)$ as

$$G(\mathcal{N}(\boldsymbol{0}, \tilde{\sigma}_2^2 \boldsymbol{I}) + \frac{1}{G}\boldsymbol{g}(D_c)), \ \text{ where } \tilde{\sigma}_2 = \sigma/G \tag{18}$$

Apparently, $SG_{q_c, \tilde{\sigma}_2}(\mathcal{T}_c) = \frac{1}{G}\boldsymbol{g}(D_c) + \mathcal{N}(\boldsymbol{0}, \tilde{\sigma}_2^2 \boldsymbol{I})$ with $\tilde{\sigma}_2 = \sigma/G$ and $q_c = L/N_c$ is also a sampled Gaussian mechanism (SGM). Since $\|\frac{1}{G}\boldsymbol{g}(D) - \frac{1}{G}\boldsymbol{g}(D')\|_2 \leq 1$, $SG_{q_c, \tilde{\sigma}_2}(\mathcal{T}_c)$ obeys $(\alpha, \Omega_{q_c, \tilde{\sigma}_2}(\alpha))$-RDP, where $\Omega_{q_c, \tilde{\sigma}_2}(\alpha) \triangleq \mathcal{D}_\alpha((1 - q_c)\mathcal{N}(0, \tilde{\sigma}_2^2) + q_c \mathcal{N}(1, \tilde{\sigma}_2^2) \| \mathcal{N}(0, \tilde{\sigma}_2^2))$.

Given the post-processing property, $\mathcal{N}(\boldsymbol{0}, \sigma^2 \boldsymbol{I}) + \boldsymbol{g}(D_c)$ also guarantees $(\alpha, \Omega_{q_c, \tilde{\sigma}_2}(\alpha))$-RDP with $q_c = L/N_c$ and $\tilde{\sigma}_2 = \sigma/G$.

According to Lemma 2.2, Algorithm 2 guarantees $(\alpha, I\Omega_{q_c, \tilde{\sigma}_2}(\alpha))$-RDP for $\mathcal{T}_c$. Since $\mathcal{T}_1, \mathcal{T}_2, ... \mathcal{T}_C$ are disjoint, based on Lemma 2.3, we know that Algorithm 2 guarantees $(\alpha, \max_c(I\Omega_{q_c, \tilde{\sigma}_2}(\alpha)))$-RDP for $\mathcal{T} = \mathcal{T}_1 \cup \mathcal{T}_2... \cup \mathcal{T}_C$.

Corollary A.1 indicates that $\max_c(I\Omega_{q_c, \tilde{\sigma}_2}(\alpha)) = I\Omega_{\max_c(q_c), \tilde{\sigma}_2}(\alpha)$. Thus, Algorithm 2 obeys $(\alpha, I\Omega_{\max_c(q_c), \tilde{\sigma}_2}(\alpha))$-RDP with $q = \max_c(L/N_c)$ and $\tilde{\sigma}_2 = \sigma/G$. ∎

Here we also prove that Lemma 2.1 is tighter than Mironov (2017)'s conversion law.

**Proof** We first provide Mironov (2017)'s conversion law:

**Lemma A.3 (RDP to DP Conversion (Mironov, 2017))** *If a randomized mechanism $\mathcal{M}$ guarantees $(\alpha, \epsilon)$-RDP, then it also obeys $(\epsilon + \log(1/\delta)/(\alpha - 1), \delta)$-DP.*

Since $(\alpha - 1)/\alpha < 1$, $\log((\alpha - 1)/\alpha) < 0$. Since $\alpha > 1$, $\log \alpha > 0$, and thus $-(\log \alpha)/(\alpha - 1) < 0$. Combining $\log((\alpha - 1)/\alpha) < 0$ and $-(\log \alpha)/(\alpha - 1) < 0$, we have

$$\log((\alpha - 1)/\alpha) - (\log \alpha)/(\alpha - 1) < 0 \ \text{ when } \alpha > 1. \tag{19}$$

We add $\epsilon + \log(1/\delta)/(\alpha - 1)$ to both sides of the above inequality and obtain

$$\epsilon + \log((\alpha - 1)/\alpha) - (\log \delta + \log \alpha)/(\alpha - 1) < \epsilon + \log(1/\delta)/(\alpha - 1) \ \text{ when } \alpha > 1. \tag{20}$$

Therefore, Lemma 2.1 is a tighter conversion law compared to Lemma A.3. ∎

## B  COMPUTATION METHOD FOR $\Omega_{q,\sigma}(\alpha)$

In this section, we briefly introduce the computation method in (Mironov et al., 2019), which is used in Meta's Opacus library (Yousefpour et al., 2021). Mironov et al. (2019) did not provide convergence analysis on the infinite series for computing the integral when $\alpha$ is a fractional number. So for completeness, we provide detailed convergence proof in this section.

Let $u(z) \triangleq \mathcal{N}(0, \sigma^2)$ and $\nu(z) \triangleq \mathcal{N}(1, \sigma^2)$, we can express $\Omega_{q,\sigma}(\alpha)$ as

$$\Omega_{q,\sigma}(\alpha) = \frac{1}{\alpha - 1} \ln \int_{-\infty}^{+\infty} u(z)(1 - q + q\frac{\nu(z)}{u(z)})^\alpha dz = \frac{1}{\alpha - 1} \ln \mathbb{E}_{z \sim u(z)}[(1 - q + q\frac{\nu(z)}{u(z)})^\alpha] \tag{21}$$

If $\alpha$ is an integer, the integral can be expressed as the sum of a finite number of terms by applying binomial expansion, *i.e.,*

$$(1 - q + q\frac{\nu(z)}{u(z)})^\alpha = \sum_{k=0}^{\alpha} (1 - q)^{\alpha - k} (q\frac{\nu(z)}{u(z)})^k = \sum_{k=0}^{\alpha} \binom{\alpha}{k} (1 - q)^{\alpha - k} q^k (\frac{\nu(z)}{u(z)})^k. \tag{22}$$

Thus, $\Omega_{q,\sigma}(\alpha)$ can be rewritten as

$$\Omega_{q,\sigma}(\alpha) = \frac{1}{\alpha - 1} \ln \mathbb{E}_{z \sim u(z)} \left[ \sum_{k=0}^{\alpha} \binom{\alpha}{k}(1-q)^{\alpha-k} q^k \left( \frac{\nu(z)}{u(z)} \right)^k \right]$$

$$= \frac{1}{\alpha - 1} \ln \sum_{k=0}^{\alpha} \binom{\alpha}{k}(1-q)^{\alpha-k} q^k \mathbb{E}_{z \sim u(z)} \left[ \left( \frac{\nu(z)}{u(z)} \right)^k \right]. \quad (23)$$

The remaining problem is to compute $\mathbb{E}_{z \sim u(z)}[(\frac{\nu(z)}{u(z)})^k]$. Given that $u(z) = \frac{1}{\sigma\sqrt{2\pi}} \exp(-\frac{z^2}{2\sigma^2})$ and $\nu(z) = \frac{1}{\sigma\sqrt{2\pi}} \exp(-\frac{(z-1)^2}{2\sigma^2})$, it is not hard to derive that

$$\mathbb{E}_{z \sim u(z)} \left[ \left( \frac{\nu(z)}{u(z)} \right)^k \right] = \exp\left( \frac{k^2 - k}{2\sigma^2} \right). \quad (24)$$

Therefore, there is an analytical solution to $\Omega_{q,\sigma}(\alpha)$ when $\alpha$ is an integer, *i.e.,*

$$\Omega_{q,\sigma}(\alpha) = \frac{1}{\alpha - 1} \ln \sum_{k=0}^{\alpha} \binom{\alpha}{k}(1-q)^{\alpha-k} q^k \exp\left( \frac{k^2 - k}{2\sigma^2} \right). \quad (25)$$

If $\alpha$ is a fractional number ($\alpha \in \mathbb{R}$), then we need to rely on the general binomial theorem to expand $(1 - q + q\frac{\nu(z)}{u(z)})^{\alpha}$.

**Lemma B.1 (General Binomial Theorem)** *If $\alpha, \gamma \in \mathbb{R}$ and $|\beta| < 1$, then $(1 + \gamma)^{\alpha}$ can be expressed as a convergent series,* i.e., $(1 + \gamma)^{\alpha} = \sum_{k=0}^{\infty} \binom{\alpha}{k} \gamma^k$, *where $\binom{\alpha}{k} = \frac{\prod_{m=0}^{k-1}(\alpha - m)}{k!}$.*

Based on Lemma B.1, we have the following corollary:

**Corollary B.1** *If $\alpha, \beta, \gamma \in \mathbb{R}$ and $|\gamma| < |\beta|$, then $(\beta + \gamma)^{\alpha}$ can also be expressed as a convergent series,* i.e., $\beta^{\alpha}(1 + \frac{\gamma}{\beta})^{\alpha} = \beta^{\alpha} \sum_{k=0}^{\infty} \binom{\alpha}{k}(\frac{\gamma}{\beta})^k = \sum_{k=0}^{\infty} \binom{\alpha}{k} \gamma^k \beta^{\alpha-k}$, *where $\binom{\alpha}{k} = \frac{\prod_{m=0}^{k-1}(\alpha - m)}{k!}$.*

Given that $z_0 = \frac{1}{2} + \sigma^2 \ln(q^{-1} - 1)$ ($z_1$ in (Mironov et al., 2019)), $1 - q > q\frac{\nu(z)}{u(z)}$ when $z < z_0$, and $1 - q < q\frac{\nu(z)}{u(z)}$ when $z > z_0$. According to Corollary B.1, we have the following convergent series:

$$\left(1 - q + q\frac{\nu(z)}{u(z)}\right)^{\alpha} = \begin{cases} \sum_{k=0}^{\infty} \binom{\alpha}{k}(1-q)^{\alpha-k} q^k \left(\frac{\nu(z)}{u(z)}\right)^k & \text{when } z < z_0 \\ \sum_{k=0}^{\infty} \binom{\alpha}{k}(1-q)^k q^{\alpha-k} \left(\frac{\nu(z)}{u(z)}\right)^{\alpha-k} & \text{when } z > z_0 \end{cases} \quad (26)$$

In the following, we prove that the two series in Eq. 26 are also convergent at $z = z_0$.

**Proof** [Eq. 26 is convergent at $z = z_0$]

If we substitute $z$ in Eq. 26 with $z_0$ ($1 - q = q\frac{\nu(z_0)}{u(z_0)}$), we can obtain

$$\sum_{k=0}^{\infty} \binom{\alpha}{k}(1-q)^{\alpha-k} q^k \left(\frac{\nu(z_0)}{u(z_0)}\right)^k = \sum_{k=0}^{\infty} \binom{\alpha}{k}(1-q)^k q^{\alpha-k} \left(\frac{\nu(z_0)}{u(z_0)}\right)^{\alpha-k} = (1-q)^{\alpha} \sum_{k=0}^{\infty} \binom{\alpha}{k}. \quad (27)$$

We first show that $\lim_{k \to +\infty} |\binom{\alpha}{k}| = 0$ by representing $|\binom{\alpha}{k}|$ as a product of three partitions, *i.e.,*

$$\left| \binom{\alpha}{k} \right| = \frac{1}{k} \cdot \left| \frac{\prod_{m=0}^{\lfloor \alpha \rfloor}(\alpha - m)}{\lfloor \alpha \rfloor!} \right| \cdot \left| \prod_{m=\lfloor \alpha \rfloor + 1}^{k-1} \frac{\alpha - m}{m} \right| \quad (28)$$

When $m \geq \lfloor \alpha \rfloor + 1$, $|\alpha - m| = m - \alpha < m$. Thus, $|\frac{\alpha - m}{m}| < 1$ when $m \geq \lfloor \alpha \rfloor + 1$. So we have $|\prod_{m=\lfloor \alpha \rfloor + 1}^{k-1} \frac{\alpha - m}{m}| < 1$. Since $|\frac{\prod_{m=0}^{\lfloor \alpha \rfloor}(\alpha - m)}{\lfloor \alpha \rfloor!}|$ is a finite number and $\lim_{k \to +\infty} \frac{1}{k} = 0$, we have $\lim_{k \to +\infty} |\binom{\alpha}{k}| = 0$.

In summary, the series $(1-q)^\alpha \sum_{k=0}^\infty \binom{\alpha}{k}$ satisfies the following conditions: (i) $\lim_{k\to+\infty} |\binom{\alpha}{k}| = 0$ (ii) When $k \geq \lfloor \alpha \rfloor + 1$ (or $k > \alpha$), $\binom{\alpha}{k}$ and $\binom{\alpha}{k+1}$ have different signs (iii) $|\binom{\alpha}{k+1}| < |\binom{\alpha}{k}|$ since $|\binom{\alpha}{k+1}|/|\binom{\alpha}{k}| = |\frac{\alpha-k}{k+1}| < 1$.

Given the above conditions, according to the alternating series test, we can conclude that $\sum_{k=0}^\infty \binom{\alpha}{k}$ is a convergent series with $\binom{\alpha}{k} = \frac{\prod_{m=0}^{k-1}(\alpha-m)}{k!}$. ∎

Therefore, we can rewrite Eq. 26 as (Eq. 12 in (Mironov et al., 2019))

$$(1 - q + q\frac{\nu(z)}{u(z)})^\alpha = \begin{cases} \sum_{k=0}^\infty \binom{\alpha}{k}(1-q)^{\alpha-k}q^k(\frac{\nu(z)}{u(z)})^k & \text{when } z \leq z_0 \\ \sum_{k=0}^\infty \binom{\alpha}{k}(1-q)^k q^{\alpha-k}(\frac{\nu(z)}{u(z)})^{\alpha-k} & \text{when } z \geq z_0 \end{cases} \tag{29}$$

As a result, we could rewrite $\Omega_{q,\sigma}(\alpha)$ as

$$\Omega_{q,\sigma}(\alpha) = \frac{1}{\alpha-1} \ln(\int_{-\infty}^{z_0} u(z) \sum_{k=0}^\infty \binom{\alpha}{k}(1-q)^{\alpha-k}q^k(\frac{\nu(z)}{u(z)})^k dz +$$

$$\int_{z_0}^{+\infty} u(z) \sum_{k=0}^\infty \binom{\alpha}{k}(1-q)^k q^{\alpha-k}(\frac{\nu(z)}{u(z)})^{\alpha-k} dz)$$

$$= \frac{1}{\alpha-1} \ln(\sum_{k=0}^\infty \binom{\alpha}{k}(1-q)^{\alpha-k}q^k \int_{-\infty}^{z_0} u(z)(\frac{\nu(z)}{u(z)})^k dz +$$

$$\sum_{k=0}^\infty \binom{\alpha}{k}(1-q)^k q^{\alpha-k} \int_{z_0}^{+\infty} u(z)(\frac{\nu(z)}{u(z)})^{\alpha-k} dz) \tag{30}$$

According to (Mironov et al., 2019), we could compute the integrals in the above series by

$$\int_{-\infty}^{z_0} u(z)(\frac{\nu(z)}{u(z)})^k dz = \frac{1}{2} \exp(\frac{k^2-k}{2\sigma^2}) \text{erfc}(\frac{k-z_0}{\sqrt{2}\sigma})$$

$$\int_{z_0}^{+\infty} u(z)(\frac{\nu(z)}{u(z)})^{\alpha-k} dz = \frac{1}{2} \exp(\frac{(\alpha-k)^2-(\alpha-k)}{2\sigma^2}) \text{erfc}(\frac{z_0-(\alpha-k)}{\sqrt{2}\sigma}). \tag{31}$$

The remaining problem is to prove that Eq. 30 is convergent. Since Mironov et al. (2019) did not provide convergence analysis, for completeness, we detail the convergence proof in the following.

**Proof** [Eq. 30 is convergent] We first prove that the first half in $\ln(\cdot)$ in Eq. 30, *i.e.,* $\sum_{k=0}^\infty \binom{\alpha}{k}(1-q)^{\alpha-k}q^k \int_{-\infty}^{z_0} u(z)(\frac{\nu(z)}{u(z)})^k dz$ is a convergent series. We rewrite the series as

$$(1-q)^\alpha \sum_{k=0}^\infty \binom{\alpha}{k}(\frac{q}{1-q})^k \int_{-\infty}^{z_0} u(z)(\frac{\nu(z)}{u(z)})^k dz \tag{32}$$

Given that $z_0 = \frac{1}{2} + \sigma^2 ln(q^{-1}-1)$, we have

$$\exp(\frac{2kz_0-k}{2\sigma^2}) = \exp(k\ln(q^{-1}-1)) = (\frac{1-q}{q})^k \tag{33}$$

Thus,

$$(\frac{q}{1-q})^k \int_{-\infty}^{z_0} u(z)(\frac{\nu(z)}{u(z)})^k dz = \frac{1}{\sigma\sqrt{2\pi}} \int_{-\infty}^{z_0} \exp\{\frac{-z^2+2k(z-z_0)}{2\sigma^2}\} dz. \tag{34}$$

Since $z - z_0 \leq 0$ for $z \in (-\infty, z_0]$, $\exp(\frac{2(z-z_0)}{2\sigma^2}) \leq 1$, and thus,

$$\frac{1}{\sigma\sqrt{2\pi}} \int_{-\infty}^{z_0} \exp\{\frac{-z^2+2(k+1)(z-z_0)}{2\sigma^2}\} dz \leq \frac{1}{\sigma\sqrt{2\pi}} \int_{-\infty}^{z_0} \exp\{\frac{-z^2+2k(z-z_0)}{2\sigma^2}\} dz. \tag{35}$$

Therefore, $(\frac{q}{1-q})^k \int_{-\infty}^{z_0} u(z)(\frac{\nu(z)}{u(z)})^k dz$ is non-increasing w.r.t. $k$. Given that $|\binom{\alpha}{k+1}| < |\binom{\alpha}{k}|$, we have one condition for the alternating convergence test, *i.e.*, $|\binom{\alpha}{k}(\frac{q}{1-q})^k \int_{-\infty}^{z_0} u(z)(\frac{\nu(z)}{u(z)})^k dz|$ is decreasing w.r.t. $k$.

Since $(\frac{q}{1-q})^k \int_{-\infty}^{z_0} u(z)(\frac{\nu(z)}{u(z)})^k = \frac{1}{\sigma\sqrt{2\pi}} \int_{-\infty}^{z_0} \exp\{\frac{-z^2 + 2k(z-z_0)}{2\sigma^2}\} dz \le \frac{1}{\sigma\sqrt{2\pi}} \int_{-\infty}^{z_0} \exp\{\frac{-z^2}{2\sigma^2}\} dz \le 1$ and $\lim_{k\to+\infty} |\binom{\alpha}{k}| = 0$, we have another condition for the alternating convergence test, *i.e.*,

$$\lim_{k\to+\infty} |\binom{\alpha}{k}(\frac{q}{1-q})^k \int_{-\infty}^{z_0} u(z)(\frac{\nu(z)}{u(z)})^k dz| = 0. \tag{36}$$

Given that $(1-q)^{\alpha-k} q^k \int_{-\infty}^{z_0} u(z)(\frac{\nu(z)}{u(z)})^k$ is positive, and when $k \ge \lfloor\alpha\rfloor + 1$, $\binom{\alpha}{k}$ and $\binom{\alpha}{k+1}$ have different signs, we know that when $k \ge \lfloor\alpha\rfloor + 1$, $\binom{\alpha}{k}(\frac{q}{1-q})^k \int_{-\infty}^{z_0} u(z)(\frac{\nu(z)}{u(z)})^k dz$ and $\binom{\alpha}{k+1}(\frac{q}{1-q})^{k+1} \int_{-\infty}^{z_0} u(z)(\frac{\nu(z)}{u(z)})^{k+1} dz$ have different signs.

Therefore, according to the alternating series test, we can conclude that the first half in $\ln(\cdot)$ in Eq. 30, *i.e.*, $(1-q)^\alpha \sum_{k=0}^{\infty} \binom{\alpha}{k}(\frac{q}{1-q})^k \int_{-\infty}^{z_0} u(z)(\frac{\nu(z)}{u(z)})^k dz$, is a convergent series.

Next, we prove that the second half in $\ln(\cdot)$ in Eq. 30, *i.e.*, $\sum_{k=0}^{\infty} \binom{\alpha}{k}(1 - q)^k q^{\alpha-k} \int_{z_0}^{+\infty} u(z)(\frac{\nu(z)}{u(z)})^{\alpha-k} dz$ is a convergent series. We rewrite the series as

$$(1-q)^\alpha \sum_{k=0}^{\infty} \binom{\alpha}{k}(\frac{q}{1-q})^{\alpha-k} \int_{z_0}^{+\infty} u(z)(\frac{\nu(z)}{u(z)})^{\alpha-k} dz \tag{37}$$

Given that $z_0 = \frac{1}{2} + \sigma^2 ln(q^{-1} - 1)$, we have

$$\exp(\frac{2(\alpha-k)z_0 - (\alpha-k)}{2\sigma^2}) = \exp((\alpha-k)\ln(q^{-1}-1)) = (\frac{1-q}{q})^{\alpha-k} \tag{38}$$

Thus,

$$(\frac{q}{1-q})^{\alpha-k} \int_{z_0}^{+\infty} u(z)(\frac{\nu(z)}{u(z)})^{\alpha-k} dz = \frac{1}{\sigma\sqrt{2\pi}} \int_{z_0}^{+\infty} \exp\{\frac{-z^2 + 2(\alpha-k)(z-z_0)}{2\sigma^2}\} dz \tag{39}$$

Since $z - z_0 \ge 0$ for $z \in [z_0, +\infty]$, $\exp(\frac{-2(z-z_0)}{2\sigma^2}) \le 1$. We then have

$$\frac{1}{\sigma\sqrt{2\pi}} \int_{z_0}^{+\infty} \exp\{\frac{-z^2 + 2(\alpha-(k+1))(z-z_0)}{2\sigma^2}\} dz =$$

$$\frac{1}{\sigma\sqrt{2\pi}} \int_{z_0}^{+\infty} \exp(\frac{-2(z-z_0)}{2\sigma^2}) \exp\{\frac{-z^2 + 2(\alpha-k)(z-z_0)}{2\sigma^2}\} dz \le$$

$$\frac{1}{\sigma\sqrt{2\pi}} \int_{z_0}^{+\infty} \exp\{\frac{-z^2 + 2(\alpha-k)(z-z_0)}{2\sigma^2}\} dz \tag{40}$$

Hence, $(\frac{q}{1-q})^{\alpha-k} \int_{z_0}^{+\infty} u(z)(\frac{\nu(z)}{u(z)})^{\alpha-k} dz$ is non-increasing w.r.t. $k$. Given that $|\binom{\alpha}{k+1}| < |\binom{\alpha}{k}|$, we have one condition for the alternating convergence test, *i.e.*, $|\binom{\alpha}{k}(\frac{q}{1-q})^{\alpha-k} \int_{z_0}^{+\infty} u(z)(\frac{\nu(z)}{u(z)})^{\alpha-k} dz|$ is decreasing w.r.t. $k$.

When $k > \alpha$, $(\frac{q}{1-q})^{\alpha-k} \int_{z_0}^{+\infty} u(z)(\frac{\nu(z)}{u(z)})^{\alpha-k} dz = \frac{1}{\sigma\sqrt{2\pi}} \int_{z_0}^{+\infty} \exp\{\frac{-z^2 + 2(\alpha-k)(z-z_0)}{2\sigma^2}\} dz \le \frac{1}{\sigma\sqrt{2\pi}} \int_{z_0}^{+\infty} \exp(\frac{-z^2}{2\sigma^2}) dz \le 1$ because $2(\alpha - k)(z - z_0) \le 0$ when $k > \alpha$. As a result, we have another condition for the alternating convergence test, *i.e.*,

$$\lim_{k\to+\infty} |\binom{\alpha}{k}(\frac{q}{1-q})^{\alpha-k} \int_{z_0}^{+\infty} u(z)(\frac{\nu(z)}{u(z)})^{\alpha-k} dz| = 0 \tag{41}$$

Since $(\frac{q}{1-q})^{\alpha-k} \int_{z_0}^{+\infty} u(z)(\frac{\nu(z)}{u(z)})^{\alpha-k} dz$ is positive, and when $k \ge \lfloor\alpha\rfloor + 1$, $\binom{\alpha}{k}$ and $\binom{\alpha}{k+1}$ have different signs, we know that when $k \ge \lfloor\alpha\rfloor + 1$, $\binom{\alpha}{k}(\frac{q}{1-q})^{\alpha-k} \int_{z_0}^{+\infty} u(z)(\frac{\nu(z)}{u(z)})^{\alpha-k} dz$ and $\binom{\alpha}{k+1}(\frac{q}{1-q})^{\alpha-(k+1)} \int_{z_0}^{+\infty} u(z)(\frac{\nu(z)}{u(z)})^{\alpha-(k+1)} dz$ have different signs.

Therefore, according to the alternating series test, we conclude that the second half in $\ln(\cdot)$ in Eq. 30, *i.e.*, $\sum_{k=0}^{\infty} \binom{\alpha}{k}(1-q)^k q^{\alpha-k} \int_{z_0}^{+\infty} u(z)(\frac{\nu(z)}{u(z)})^{\alpha-k} dz$ is a convergent series.

All in all, Eq. 30 is convergent. ∎

The practical computation of $\Omega_{q,\sigma}(\alpha)$ when $\alpha$ is a fractional number proceeds by computing $\int_{-\infty}^{z_0} u(z)(\frac{\nu(z)}{u(z)})^k dz$ and $\int_{z_0}^{+\infty} u(z)(\frac{\nu(z)}{u(z)})^{\alpha-k} dz$ with Eq. 31 and then plugging in the results into the series Eq. 30 until convergence (Mironov et al., 2019).

## C  SYNTHETIC DATA DEFINED BY EQ. 8

According to (Dong et al., 2022), the synthetic data defined by Eq. 7 is represented under the orthogonal basis $\mathcal{E} = \{e_1, e_2, ..., e_d\}$. Thus, we need to transform the synthetic data back to the standard basis with Eq. 8. Here we explain how to compute the synthetic dataset defined by Eq. 8. For each class $c$, we flatten and concatenate the data from $\mathcal{T}_c$ into a data matrix $X_c$ with $|\mathcal{T}_c|$ rows and $d$ columns. Each row of $X_c$ is a data sample with dimension $d$. We then compute the transformation matrix by QR decomposition, *i.e.*, $X_c = Q_c \tilde{X}_c$. After that, we randomly sample $M$ Gaussian noise samples $\{\tilde{s}_{c,i}\}_{i=1}^{M}$ from $\mathcal{N}(\mathbf{0}, I_d)$ and compute $M$ synthetic samples for class $c$ by Eq. 8, *i.e.*, $s_{c,i}^{*} = Q_c \tilde{s}_{c,i} + \frac{1}{|\mathcal{T}_c|} \sum_{j=1}^{|\mathcal{T}_c|} x_{c,j}$, where $\mathcal{T}_c = \{x_{c,j}, c\}_{j=1}^{|\mathcal{T}_c|}$ and $x_{c,j}$ is also flattened. Finally, we reshape $s_{c,i}^{*}$ into the original data shape. The synthetic dataset is $\{\{s_{c,i}^{*}\}_{i=1}^{M}\}_{c=1}^{C}$.

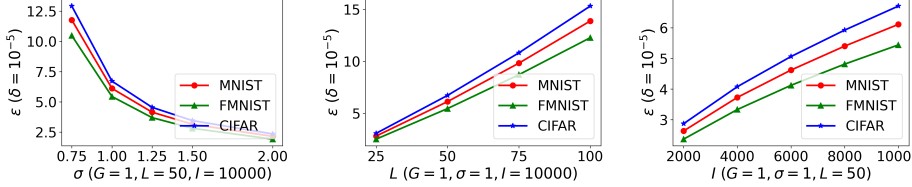

Figure 1: The privacy budgets of NDPDC with different $\sigma$, $L$, and $I$.

| Dataset | Noise | Test Acc | DP budget |
|---------|-------|----------|-----------|
| MNIST | $\sigma = 2$ | $96.46\% \pm 0.20\%$ | $(2.16, 10^{-5})$-DP |
| | $\sigma = 3$ | $95.89\% \pm 0.07\%$ | $(1.32, 10^{-5})$-DP |
| | $\sigma = 4$ | $95.59\% \pm 0.20\%$ | $(0.95, 10^{-5})$-DP |
| | $\sigma = 5$ | $94.96\% \pm 0.19\%$ | $(0.74, 10^{-5})$-DP |
| FMNIST | $\sigma = 2$ | $80.96\% \pm 0.39\%$ | $(1.93, 10^{-5})$-DP |
| | $\sigma = 3$ | $79.43\% \pm 0.46\%$ | $(1.19, 10^{-5})$-DP |
| | $\sigma = 4$ | $78.42\% \pm 0.41\%$ | $(0.85, 10^{-5})$-DP |
| | $\sigma = 5$ | $77.81\% \pm 0.39\%$ | $(0.67, 10^{-5})$-DP |
| CIFAR10 | $\sigma = 2$ | $47.93\% \pm 0.43\%$ | $(2.36, 10^{-5})$-DP |
| | $\sigma = 3$ | $45.03\% \pm 0.62\%$ | $(1.44, 10^{-5})$-DP |
| | $\sigma = 4$ | $42.73\% \pm 0.54\%$ | $(1.04, 10^{-5})$-DP |
| | $\sigma = 5$ | $41.20\% \pm 0.51\%$ | $(0.81, 10^{-5})$-DP |
| CelebA | $\sigma = 2$ | $78.49\% \pm 0.42\%$ | $(0.17, 10^{-5})$-DP |
| | $\sigma = 3$ | $77.38\% \pm 1.05\%$ | $(0.09, 10^{-5})$-DP |
| | $\sigma = 4$ | $76.46\% \pm 1.32\%$ | $(0.07, 10^{-5})$-DP |
| | $\sigma = 5$ | $76.22\% \pm 0.75\%$ | $(0.05, 10^{-5})$-DP |

Table 8: The performance of NDPDC with a variety of DP budgets.

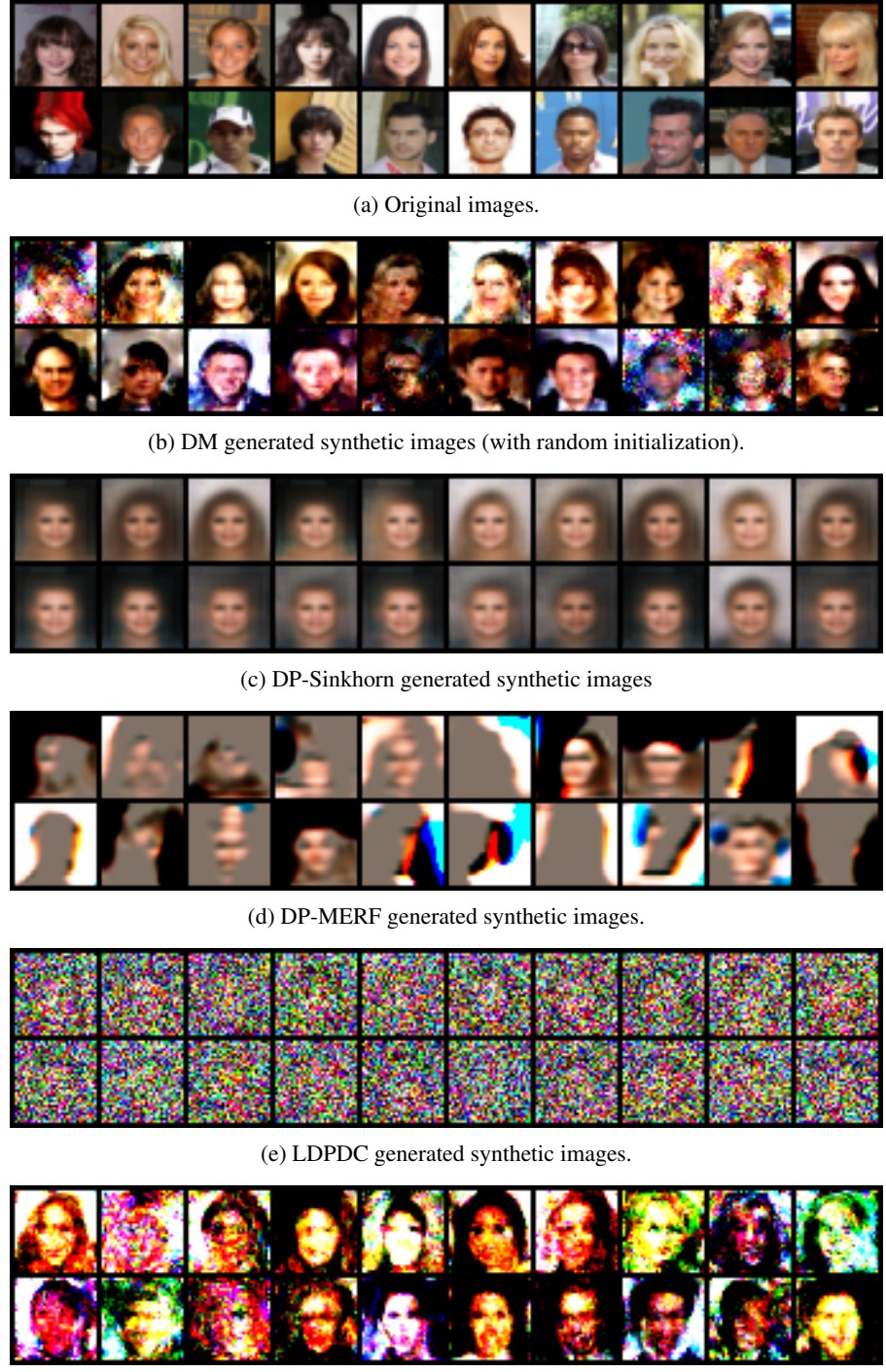

(a) Original images.

(b) DM generated synthetic images (with random initialization).

(c) DP-Sinkhorn generated synthetic images

(d) DP-MERF generated synthetic images.

(e) LDPDC generated synthetic images.

(f) NDPDC generated synthetic images.

Figure 2: Visualizing the synthetic CelebA images. Female synthetic images are listed in the first row, and male synthetic images are listed in the second row.

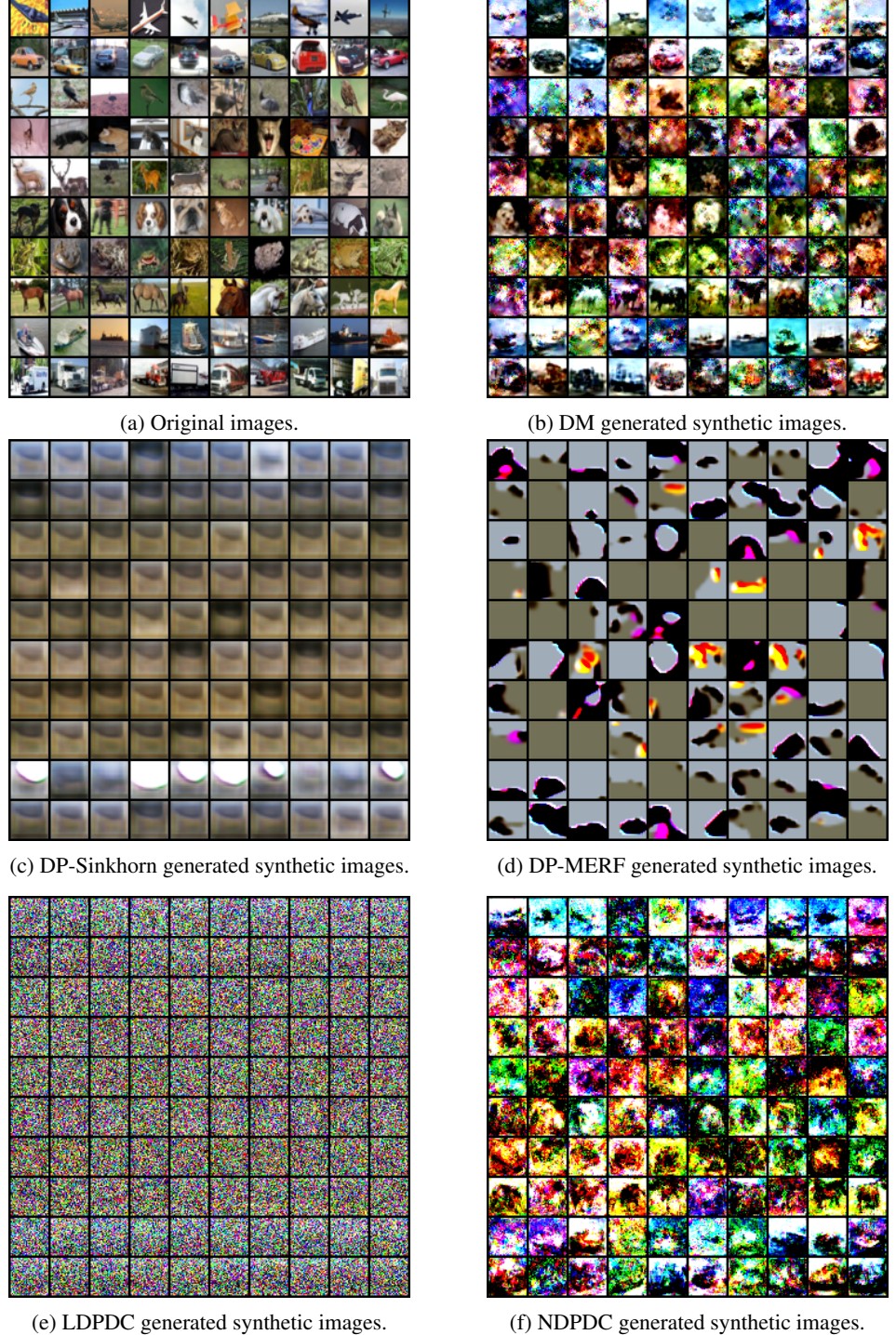

(a) Original images.

(b) DM generated synthetic images.

(c) DP-Sinkhorn generated synthetic images.

(d) DP-MERF generated synthetic images.

(e) LDPDC generated synthetic images.

(f) NDPDC generated synthetic images.

Figure 3: Visualizing the synthetic CIFAR10 images.

## D ADDITIONAL EXPERIMENTAL RESULTS

We visualize synthetic images generated by DM, DP-Sinkhorn, DP-MERF, LDPDC, and NDPDC in Fig. 2 and Fig. 3. Compared to DM-generated images, NDPDC-generated images are more noisy due to the DP guarantees. A surprising result is that LDPDC-generated images look like noise but the models still can learn some information from LDPDC-generated images. We conjecture that this is because LDPDC-generated images still have certain patterns, but the patterns are hardly perceptible by human beings because of the high noise ($\sigma = \sqrt{d}$).

We remark that, although the synthetic images generated by DP-Sinkhorn on CelebA look like faces, they are not very colorful and not diverse. Therefore, when being tested on the colorful and diverse original CelebA images, the model trained on NDPDC-generated images has better accuracy than the model trained on DP-Sinkhorn generated images.

| Dataset | Test Acc | DP budget |
|---------|----------|-----------|
| MNIST | $86.70\% \pm 2.07\%$ | $(11.60, 10^{-5})$-DP |
| FMNIST | $70.38\% \pm 0.79\%$ | $(11.60, 10^{-5})$-DP |
| CIFAR10 | $20.61\% \pm 0.87\%$ | $(11.60, 10^{-5})$-DP |
| CelebA | $69.51\% \pm 1.69\%$ | $(11.60, 10^{-5})$-DP |

Table 9: The performance of DP-MERF with $\sigma = 0.5$. We employ 50 synthetic samples per class to train the ConvNet models for evaluation.

Beyond the visualization results, we report the results of NDPDC with high $\sigma$ and low DP budgets in Table 8. We set $\sigma$ set to $0.5$ and report the results of DP-MERF (Harder et al., 2021) in Table 9. We also evaluate DP-HP using Vinaroz et al. (2022)'s publicly available code** and report the results on MNIST and FMNIST in Table 10.

| Dataset | Test Acc | DP budget |
|---------|----------|-----------|
| MNIST | $74.20\% \pm 1.66\%$ | $(1, 10^{-5})$-DP |
| FMNIST | $28.05\% \pm 1.12\%$ | $(1, 10^{-5})$-DP |

Table 10: The performance of DP-HP. We follow the instructions and run the code from `https://github.com/ParkLabML/DP-HP` to generate synthetic data. We employ 50 synthetic samples per class to train the ConvNet models for evaluation.

## E EXTENDED RELATED WORK

**Generative Methods**   The previous literature has proposed an array of generative methods for synthetic data generation (Kingma & Welling, 2013; Goodfellow et al., 2014; Mirza & Osindero, 2014; Higgins et al., 2016; Arjovsky et al., 2017; Brock et al., 2018). Due to the growing privacy concern, recent research also focuses on developing differentially private generative methods (Xie et al., 2018; Jordon et al., 2018; Torkzadehmahani et al., 2019; Long et al., 2021). Xie et al. (2018) first combined DP-SGD and GAN to generate private synthetic data. Torkzadehmahani et al. (2019) combined conditional GAN and DP-SGD to generate class-conditional private data. Jordon et al. (2018) applied PATE (Papernot et al., 2016) to GAN and developed a differentially private GAN framework called PATE-GAN. PATE-GAN trains a student discriminator on the labels output by the PATE mechanism and trains the generator on the generative loss computed over the student discriminator. (Long et al., 2021) proposed a framework called G-PATE with a private gradient aggregation mechanism to enable a better combination of PATE and GAN. GS-WGAN (Chen et al., 2020) proposed to selectively apply the randomized mechanism in DP-SGD to maximally preserve the true gradient direction and use the Wasserstein objective to improve the amount of gradient information flow during training the generative models. DP-MERF (Harder et al., 2021) proposed to train the generator by matching the mean embeddings of the real data and the generator-output

---

** `https://github.com/ParkLabML/DP-HP`

synthetic data. The main differences between DP-MERF and NDPDC include (i) NDPDC uses neural network based feature extractors to compute the representations, while DP-MERF uses random Fourier features to compute the embeddings; (ii) NDPDC directly optimizes on the synthetic data, while DP-MERF optimizes the generative model parameters; (iii) NDPDC and DP-MERF apply the Gaussian mechanism in a different way. DP-Shinkhorn (Cao et al., 2021) framed the generative learning problem as minimizing the optimal transport distance and trained the generative models using a semi-debiased Sinkhorn loss. Cao et al. (2021) demonstrated that, using $(10, 10^{-5})$-DP budget, DP-Shinkhorn can generate synthetic data with better utility and quality than G-PATE and GS-WGAN on MNIST and FashionMNIST.

**Differential Privacy**    In addition to DP and RDP, the prior research has proposed some other differential privacy notations, such as Concentrated Differential Privacy (CDP) (Dwork & Rothblum, 2016), zero CDP (Bun & Steinke, 2016), and truncated CDP (Bun et al., 2018). CDP (Dwork & Rothblum, 2016) is a relaxation of DP—An algorithm obeys $(\mu, \tau)$-CDP if privacy loss random variable has mean $\mu$, and its deviation from $\mu$ is subgaussian with standard $\tau$. Zero CDP (Bun & Steinke, 2016) is an alternative formulation of CDP, and truncated CDP is a relaxation of zero CDP. Beyond proposing new notations for enhancing privacy analysis, the prior literature also studied privacy amplification by (sub)sampling, which was first proposed in (Kasiviswanathan et al., 2011; Beimel et al., 2013). The randomness introduced by sampling benefits the analysis in (Bassily et al., 2014; Foulds et al., 2016). Recently, Wang et al. (2019); Mironov et al. (2019) studied the (sub)sampled Gaussian mechanism—a combination of subsampling and the Gaussian mechanism and delivered privacy analysis under the framework of RDP. We follow Opacus (Yousefpour et al., 2021) to exploit (Mironov et al., 2019)'s results in our proof and experiments and provide detailed convergence analysis for (Mironov et al., 2019)'s computation method.

## F  ADDITIONAL DISCUSSION

**Additional Discussion on LDPDC and DPMix**    Similar to LDPDC, DPMix (Lee et al., 2019) is a linear algorithm for differentially private data generation. However, there are some differences between LDPDC and DPMix:

1. LDPDC does not need to randomize the labels with the help of the parallel composition law, but DPMix needs to randomize the labels. We note that adding noise to the labels may hurt the model performance.

2. LDPDC adds noise to the sum of samples and divides it by the fixed group size $L$, while DPMix directly adds noise to the mean of the samples. For LDPDC, dividing the randomized result by the group size $L$ could help dilute the negative effects of random noise on model performance. Actually, we find that, for LDPDC, if we directly add noise to mean of samples like DPMix, the performance of LDPDC will become much worse.

3. DPMix uses sampling without replacement, while LDPDC uses Poisson sampling. Note that Poisson sampling is the standard sampling method used in the state-of-the-art Pytorch library for differentially private deep learning (Yousefpour et al., 2021).

We have reproduced DPMix and observe that LDPDC has better performance than DPMix with the settings for dataset condensation. Specifically, we set $\sigma_X = \sigma_Y = 1$, $L = 50$, and $M = 50$, the result that we reproduce for DPMix is only $10.22\% \pm 1.23\%$ on CIFAR10. We conjecture that this may be because (i) The operations of adding noise to the labels and the mean of the samples in DPMix cause more negative effects on model performance, compared to LDPDC. (ii) DPMix may be only able to use a large number of synthetic samples to achieve the results reported in (Lee et al., 2019). (iii) There are some missing details in the published version of (Lee et al., 2019) that may affect the effectiveness of DPMix. If we instead refer to the results reported in (Lee et al., 2019), we also observe that LDPDC can use lower DP budgets to achieve comparable performance to DPMix.

**General DP Data Generation vs. DP Dataset Condensation**    There is a major difference between general DP data generation methods and DP dataset condensation methods: General DP data generation methods typically target at training generative models to generate new synthetic data samples, while DP dataset condensation methods aim to condense the original dataset into a small

synthetic dataset and maintain the data utility for training models. To achieve the respective goals, DP generative methods optimize the generative model parameters, while NDPDC directly optimizes the small synthetic dataset. Thus, DP data generation methods may be more useful for generating new data samples with the learned generators, but DP dataset condensation is more useful for data-efficient learning (Zhao & Bilen, 2021a; Zhao et al., 2021; Zhao & Bilen, 2021b). This is because DP dataset condensation significantly reduces the cost of data storage and model training, and the models trained on a small amount of data produced by DP dataset condensation methods have better utility than the models trained on a small amount of data generated by DP generative methods.

**Practical Use of DP Dataset Condensation**   A useful and practical application of DP dataset condensation methods is: Data owners with privacy concerns could condense their datasets into small synthetic datasets by DP dataset condensation methods. With DP protection, the data owners would feel more comfortable to share the synthetic datasets with other users or devices. Even if some users or devices do not have much computational resource or storage, they could store the small synthetic datasets and train models with not bad utility. The data owners could also share their data with trusted entities such as governments, hospitals, or banks. To mitigate the data owners' concerns, the trusted entities can execute DP dataset condensation methods on the collected data and share the small synthetic datasets with other users or devices. In this application, DP dataset condensation is a better choice than general DP data generation methods since the models trained on the small synthetic datasets produced by DP dataset condensation methods have better utility than the models trained on the data generated by DP generation methods.

