# OpenReview forum: "Differentially Private Dataset Condensation"
_ICLR.cc/2023/Conference — Submitted to ICLR 2023_

### Official Review · Reviewer_Spfc · 2022-10-25

**Confidence:** 4
**Clarity, Quality, Novelty And Reproducibility:** See the *Strength And Weaknesses*.
**Correctness:** 4
**Technical Novelty And Significance:** 2
**Empirical Novelty And Significance:** 2
**Recommendation:** 6

**Strength And Weaknesses:**

**Strengths**
1. The experiment results of two proposed methods show the promising improvement from the baselines.

**Weakness**
1. My major concern is that this paper misses an important and relevant literature [1], which might influence the novelty and significance of the results. The algorithm in [1] is similar to Algorithm 1 in this paper. The slight difference is that in this paper, the data batch is sampled by Poisson sampling, while [1] samples the data without replacement. Moreover, it is questionable how much improvement LDPDC and NDPDC have by comparing them with this literature.
2. The main experiment results could be more convincible if different algorithms can be compared with the (mostly) fixed DP-Budget. It is also worth showing at which $epsilon$ the performance of NDPDC would break down. They will help understand the comparison with the baselines. They are also convenient for later work to build the comparison with this work.
3. The writing of this paper can be improved. Below some clarity questions or suggestions are listed:
 - The abbreviation "DM" is not introduced.
 - The definition of $T_c$ is introduced in page 5, but appears first in page 4.
 - The number of iteration $I$ is not mentioned in the Algorithm 2.
 - Missing the description of whether the feature extractor $\phi$ is pretrained.
 - Missing the description of data augmentation in Algorithm 2.
 - What is the privacy budget setting for Table 2
 - Which column is the "ConvNet" mentioned in the second paragraph of section 4.2

[1] Lee, Kangwook, et al. "Synthesizing differentially private datasets using random mixing." 2019 IEEE International Symposium on Information Theory (ISIT). IEEE, 2019.


--------After Discussion-------
Given that the difference between [1] and LDPDC is clarified, I raised my score to 6. I recommend authors to add the discussion and solve the clarification questions in the revision.

**Summary Of The Paper:**

The paper first discusses the issues in the controversial assumptions in a recent literature. Then it proposes two differentially private dataset condensation algorithms LDPDC and NDPDC. In the experiment, it makes the evaluations on multiple datasets and the results show that the proposed two methods achieves better privacy-utility trade-off than baselines.

**Summary Of The Review:**

I appreciate this work studies the dataset condensation with differential privacy. However, because of the missing discussion and comparison with [1], the novelty and result significance are doubt. Therefore, I recommend reject.

---

> ### Author Response · Authors · 2022-11-05
> **We are sorry that [1] does not provide some essential details for us to reproduce more results on CNN. (We eagerly look forward to your valuable feedback).**
>
> We tried to reproduce [1] (for providing more results of [1] on CNN), but unfortunately, **[1] does not provide some essential details for us to reproduce it.** The paper that we refer to is the published version of [1] from https://ieeexplore.ieee.org/abstract/document/8849381 The paper also says another version can be accessed at http://csuh.kaist.ac.kr/ISIT2019_DPMix_full.pdf (But we could **not** open this link).
>
> The missing details include but are not limited to
>
> (1) The loss function used for training nonlinear models?
>
> Since [1] randomizes the label by Gaussian noise, the randomized label is a vector, which is **not** a probability vector since the sum of the vector is not 1. Then, a natural question is what loss function we should use for [1]? *We tried MSE loss and trained a CNN on CIFAR10, but the result is very bad (close to random guess)*. We also tried to apply softmax function to the label vectors and use cross-entropy loss, but the result is also not satisfactory.
>
> (2) The CNN architecture?
>
> [1] says that it uses a small CNN for experiments but does not provide any detail about the CNN architecture.
>
> (3) Other technical details for reproducing the results in Table 1 and 2 in [1]?
>
> We did not find the optimization details (like the learning rate, the number of steps) used for training the CNN. We did not see the $\sigma_x$ and $\sigma_y$ used for reproducing the results in Table 1 and 2. We also did not find the number of synthetic samples used for training. We only know that [1] sets the T (seems to be the number of synthetic samples) as $10^6$ for Fig. 3. This setting is not suitable for dataset condensation since for dataset condensation, a normal setting is 50 synthetic samples per class.
>
> **Since [1] does not release its code and misses some essential technical details about experiments on CNN, we currently could not reproduce [1]. Please let us know if you know more details about [1], or you could open http://csuh.kaist.ac.kr/ISIT2019_DPMix_full.pdf**
>
> **Therefore, at the current stage, we could only refer to the results reported in [1]. According to Table 1 and 2 in [1], it is very obvious that NDPDC has much better performance than [1].**
>
> We eagerly look forward to your valuable feedback. If you do not have other major concerns, we hope you could consider raising the score. If you have other major concerns, please let us know the concerns, and we will work on addressing those concerns.
>
> Reference
>
> [1] Lee, Kangwook, et al. "Synthesizing differentially private datasets using random mixing." 2019 IEEE International Symposium on Information Theory (ISIT). IEEE, 2019.

---

> ### Author Response · Authors · 2022-11-07
> **A detailed response to address your concerns and questions. (We eagerly look forward to your valuable feedback).**
>
> We appreciate your valuable time to review this paper. We provide a detailed response to your concerns and questions below. **We eagerly look forward to your valuable feedback on our response.**
>
> > Q1. Missing comparison with [1]. It is questionable how much improvement LDPDC and NDPDC have by comparing them with [1].
>
> A: **NDPDC is completely different from [1], and NDPDC has much better performance than [1]. [1] is a linear algorithm, while NDPDC is a nonlinear algorithm that is more suitable for nonlinear problems such as image recognition. According to the results reported in [1], it is obvious that NDPDC has much better performance than [1], as illustrated in the table below. The improvement is very clear.** We believe that NDPDC and the theoretical results are our main contributions. We present LDPDC in the paper mainly because it is very simple to implement and easy to use. Also, LDPDC can use low privacy budgets to achieve comparable performance to DP Sinkhorn, DP MERF, and [1].
>
>
> |            |            MNIST                         |  CIFAR10                             |
> |:-------:|:------------------------------------:|:------------------------------------:|
> |  NDPDC  |  95.32\%$\pm$0.29\% ($\epsilon=1$)  | 42.40\%$\pm$0.86\% ($\epsilon=1$)     |
> |  NDPDC  |  97.35\%$\pm$0.13\% ($\epsilon=6.12$)| 52.68\%$\pm$0.40\% ($\epsilon=6.72$) |
> |   [1]   |      80.6\% ($\epsilon=20$)          |       28.9\% ($\epsilon=30$)         |
>
> **We really appreciate the work done by [1]. However, since [1] does not release its code, and its published version does not provide some essential technical details for us to reproduce the results on CNN.** (See our previous response for more details). So at the current stage, we are sorry that we could mainly refer to the results reported in [1].
>
> **In the revision, we will mention [1], and we will add a statement that that LDPDC has comparable performance to [1], but NDPDC has much better performance than [1] since NDPDC is a nonlinear algorithm that is more suitable for nonlinear problems such as image recognition. Does this statement look good to you? Please let us know your valuable feedback.**
>
> > Q2: The main experiment results could be more convincible if different algorithms can be compared with the (mostly) fixed DP-Budget. It is also worth showing at which $\epsilon$ the performance of NDPDC would break down. They will help understand the comparison with the baselines. They are also convenient for later work to build the comparison with this work.
>
> A: We appreciate this insightful comment. We use different budgets in Table 1 because different papers' default settings use different budgets, e.g., DP-Sinkhorn uses
> $\epsilon=10$ but DP-MERF uses $\epsilon=1$. We conduct experiments under $\epsilon=1$ and provide results in the following table. For all the methods, we use number_classes * 50 synthetic samples for training the models, and test the models on the real testing datasets. It is clear that NDPDC has much better performance than the other methods.
>
> | $\epsilon=1$ |        MNIST       |        FMNIST      |       CIFAR10      | CelebA (2 classes) |
> |:------------:|:------------------:|:------------------:|:------------------:|:------------------:|
> |    LDPDC     | 85.80\%$\pm$0.39\% | 63.64\%$\pm$0.76\% | 25.46\%$\pm$0.83\% | 69.64\%$\pm$1.01\% |
> |    NDPDC     | 95.32\%$\pm$0.29\% | 78.79\%$\pm$0.37\% | 42.40\%$\pm$0.86\% | 81.47\%$\pm$0.80\% |
> | DP Sinkhorn  | 55.43\%$\pm$1.54\% | 43.22\%$\pm$1.40\% | 12.62\%$\pm$0.27\% | 64.02\%$\pm$0.48\% |
> |   DP MERF    | 84.81\%$\pm$2.04\% | 63.05\%$\pm$2.05\% | 17.10\%$\pm$0.78\% | 69.26\%$\pm$0.90\% |
>
> **Under reasonable settings of $\epsilon$ ($1\leq \epsilon \leq 10$), we do not observe a severe breakdown of NDPDC. Even with $\epsilon=1$, NDPDC achieves much better performance than DP MERF, DP Sinkhorn, and [1].** We think $1\leq \epsilon \leq 10$ is a reasonable range since the privacy guarantee with $\epsilon > 10$ is considered not very useful by some researchers in the community, and it is rare to see $\epsilon < 1$ in the previous works.
>
> **It is very easy for the following works to compare with our work because we provide our code in the supplementary material, and we will release the code on Github.** The "compute_sigma_with_fixed_budget.py" in the supplementary material can help compute the $\sigma$ required for LDPDC and NDPDC with a predefined $\epsilon$.
>
>
> > Q3 and our answer are given below (due to the limited space).

---

> > ### Author Response · Authors · 2022-11-07
> > **The second part of the detailed response.**
> >
> > > Q3. The writing of this paper can be improved. Below some clarity questions or suggestions are listed:
> >
> > We sincerely thank you for your comments that help us improve the clarity of the paper.
> >
> > (1) The abbreviation "DM" is not introduced.
> >
> > "DM" refers to distribution matching [2]. We will explain it right after the introduction of distribution matching in Section 2.1 in the revision.
> >
> > (2) The definition of $\mathcal{T}_c$ is introduced in page 5, but appears first in page 4.
> >
> > We will move the footnote regarding $\mathcal{T}_c$ from Page 4 to Page 5 in the revision.
> >
> > (3) The number of iteration $I$ is not mentioned in the Algorithm 2.
> >
> > The number of iterations $I$ means the number of iterations to execute the iterative procedure (for each iteration do) in Algorithm 2. From our perspective, it seems clear. We will include the number of iteration $I$ in the requirement of Algorithm 2.
> >
> > (4) Missing the description of whether the feature extractor $\{\Phi}_{\mathbf{\mathit{\theta}}}$ is pretrained.
> >
> > In Algorithm2, we mentioned that we (randomly) sample $\mathbf{\mathit{\theta}}$, so it is not pretrained, which is the same as [2]. We did not expect it to cause ambiguity, and we will clarify it in the revision.
> >
> > (5) Missing the description of data augmentation in Algorithm 2.
> >
> > In the paper, we mention that we follow [3] to use data augmentation. [2] also follows [3] to use data augmentation. We do not have space in the main context to introduce the details of data augmentation, and we expect interested readers to refer to [3] or our code for more details.
> >
> > (6) What is the privacy budget setting for Table 2.
> >
> > For Table 2, we just use the default settings (Section 4.1), so the budgets are given in Table 1. We will mention this in the caption of Table 2.
> >
> > (7) Which column is the "ConvNet" mentioned in the second paragraph of section 4.2.
> >
> > The results regarding ConvNet are in Table 1. For Table 1 and 2, we use the default settings (Section 4.1) for LDPDC and NDPDC, and we will mention this in the revision.
> >
> > **If we have addressed your major concern, we hope you will consider raising the score. We eagerly look forward to your valuable feedback. Thanks again for your valuable review.**
> >
> > [1] Lee, Kangwook, et al. "Synthesizing differentially private datasets using random mixing." 2019 IEEE International Symposium on Information Theory (ISIT). IEEE, 2019.
> >
> > [2] Bo Zhao and Hakan Bilen. Dataset condensation with distribution matching. CoRR, abs/2110.04181, 2021
> >
> > [3] Bo Zhao and Hakan Bilen. Dataset condensation with differentiable siamese augmentation. In International Conference on Machine Learning, pp. 12674–12685. PMLR, 2021

---

> ### Author Response · Authors · 2022-11-16
> **Could you please check our response? Thank you in advance for your kind cooperation!**
>
> Dear Reviewer Spfc,
>
> Since there are only three days left in the discussion period, we would greatly appreciate it if you could provide some feedback on our response and revision soon. We believe we have addressed your major concern.
>
> > It is questionable how much improvement LDPDC and NDPDC have by comparing them with [1]?
>
> According to the results reported in [1], NDPDC outperforms [1] by 13\%-24\% in testing accuracy on MNIST and CIFAR10. This is because [1] is a linear algorithm, while NDPDC is a nonlinear algorithm that is more suitable for nonlinear problems such as image recognition. **We have provided more details in the previous response.**
>
> In the revision, we add discussion on [1] and NDPDC in Section 4.2 (also marked by the blue color).
>
> If our response addresses your major concern, please consider raising the score.
> Thank you very much for your time to review this paper!
>
> [1] Lee, Kangwook, et al. "Synthesizing differentially private datasets using random mixing." 2019 IEEE International Symposium on Information Theory (ISIT). IEEE, 2019.

---

> > ### Comment · Reviewer_Spfc · 2022-11-16
> > **Comparison with Lee, Kangwook, et al.**
> >
> > Thank authors for the reply. I understand there would be some improvement from [1] to NDPDC, but I am still concerned about the novelty/improvement compared with LDPDC. Here are some detailed questions about the comparison:
> > 1. In the aspect of the algorithm, the only difference is that [1] samples each subset without replacement but LDPDC samples each subset by Poisson sampling. Different sampling may lead to difference analysis of privacy. When $(\ell, \sigma_X, T)$ in [1] are the same as $(L, \sigma, M)$ in LDPDC, does LDPDC cost smaller privacy budget?
> > 2. For the empirical evaluation, one can run [1] with $\ell=L$, $\sigma_X=\sigma$ and synthetic dataset size $T=M$. Then train the same classifiers for both [1] and LDPDC. With this set-up, does LDPDC have better utility-privacy tradeoff than [1]?
> >
> > [1] Lee, Kangwook, et al. "Synthesizing differentially private datasets using random mixing." 2019 IEEE International Symposium on Information Theory (ISIT). IEEE, 2019.

---

> > > ### Author Response · Authors · 2022-11-16
> > > **Thank you for your comments, and we eagerly look forward to your feedback.**
> > >
> > > We greatly appreciate your reply!
> > >
> > > We note that **NDPDC and the extensive theoretical and empirical analysis are the most important contributions of this paper.** **The improvement of NDPDC over [1] is significant**. We include LDPDC in this paper mainly because it is simple to use (with our code) and has comparable performance to recent DP generative methods. **So the novelty of LDPDC should *not* be the only reason for directly rejecting this paper.**
> > >
> > > > Q1 In the aspect of the algorithm, the only difference is that [1] samples each subset without replacement but LDPDC samples each subset by Poisson sampling.
> > >
> > > Sampling is **actually not** the only difference between LDPDC and DPMix [1]. The other differences include:
> > >
> > > (1) LDPDC does not need to randomize the labels (please check our proof to see the reason, thanks to the parallel composition law), but [1] needs to randomize the labels.
> > >
> > > (2) LDPDC adds noise to the sum of samples and divides it by the fixed group size, while [1] adds noise to the mean of the samples. For LDPDC, dividing the noise + sum by the group size could help dilute the negative effects of random noise. **According to the results reported in [1], LDPDC indeed uses much lower privacy budgets to achieve comparable performance to [1].**
> > >
> > > >  Q2 For the empirical evaluation, one can run [1] with $\ell=L$, $\sigma_X=\sigma$ and synthetic dataset size $T=M$ Then train the same classifiers for both [1] and LDPDC. With this set-up, does LDPDC have better utility-privacy tradeoff than [1]?
> > >
> > > **As we mentioned in the previous response, we tried to reproduce [1] with our settings (e.g., $T=M$), but the results of [1] are very bad (close to random guess on CIFAR10).**
> > >
> > > **One reason may be that the published version of [1] could not provided enough technical details due to its limited space, and [1] does not release its code.** As a result, we face a lot of problems in the process of reproducing [1]. For instance, one problem that we face is which loss function we should use for [1] on CNN. [1] randomizes the one-hot labels, and the randomized one-hot labels are even not probability vectors. **We tried MSE and $T=M$ for [1] but got bad results.** So what loss function should we use for [1] on CNN? Even if we know the loss function, there are other missing details in the published version of [1], as we mentioned in the previous response.
> > >
> > > Another reason may be that [1] uses a very large $T$ to achieve the results reported in [1]. In contrast, $M$ is very small (e.g., only 50). So with such a small $M$, the accuracy achieved by [1] will significantly decrease, especially on complicated and nonlinear datasets such as CIFAR10.
> > >
> > > **We re-emphasize that NDPDC and the extensive theoretical and empirical analysis are the most important contributions of this paper.** The improvement of NDPDC over [1] is significant.  So we respectfully disagree that the novelty of LDPDC could be the only reason for directly rejecting this paper.**
> > >
> > > Thank you for your comments, and we eagerly look forward to your feedback.
> > >
> > > [1] Lee, Kangwook, et al. "Synthesizing differentially private datasets using random mixing." 2019 IEEE International Symposium on Information Theory (ISIT). IEEE, 2019.

---

> > > > ### Author Response · Authors · 2022-11-16
> > > > **Our reproduced results of [1] (we eagerly look forward to your valuable feedback).**
> > > >
> > > > We greatly appreciate your reply!
> > > >
> > > > Since [1] does not release it code, and the published version of [1] (only 5 pages with references) does not have enough space to provide enough details, we could only reproduce [1] with limited information.
> > > >
> > > > We set $\sigma_X=\sigma_Y=1$, $L=50$, and $M=50$ (number of samples per class). The result we got for [1] is only 10.22\% $\pm$ 1.23\% on CIFAR10. The reasons for this result may be:
> > > >
> > > > 1. T is very large, and M is small. [1] may be only able to use large T to achieve the results reported in [1].
> > > >
> > > > 2. [1] needs to randomize the labels. Randomizing the one-hot labels may ruin the performance. One advantage of LDPDC over [1] is that it does not need to randomize the labels (please check our proof to see the reason).
> > > >
> > > > 3. There are some missing details in the published version of [1] that may affect the effectiveness of [1] on CNN.
> > > >
> > > > [1] Lee, Kangwook, et al. "Synthesizing differentially private datasets using random mixing." 2019 IEEE International Symposium on Information Theory (ISIT). IEEE, 2019.

---

> ### Author Response · Authors · 2022-11-16
> **Please let us know if we address the only remaining concern. Thanks for your kind cooperation!**
>
> Since there are only two days left in the discussion period, we would greatly appreciate it if you could provide us with more feedback. We believe we have addressed your *major* concern with the following response.
>
> > Novelty of LDPDC
>
> NDPDC and the extensive theoretical and empirical analysis are the most important contributions of this paper. The improvement of NDPDC over [1] is significant. We include LDPDC in this paper mainly because it is simple to use (with our Python code) and has comparable performance to recent DP generative methods. So the novelty of LDPDC should **not** be a major concern for this paper, and should **not** be the only reason for directly rejecting this paper.
>
>
> > In the aspect of the algorithm, the only difference is that [1] samples each subset without replacement but LDPDC samples each subset by Poisson sampling.
>
> LDPDC and [1] have other differences:
>
> (1) LDPDC does not need to randomize the labels (please check our proof to see the reason, thanks to the parallel composition law), but [1] needs to randomize the labels. Adding noise to the labels could hurt the model performance.
>
> (2) LDPDC adds noise to the sum of samples and divides it by the fixed group size, while [2] adds noise to the mean of the samples. For LDPDC, dividing the noise + sum by the group size could help dilute the negative effects of random noise. **We find that, even for LDPDC, if we directly add noise to mean of samples like [1], the performance of LDPDC will also become much worse.**
>
> > Comparison with [1] (Comparing LDPDC with [1])
>
> (1) NDPDC performs much better than [1]. On CIFAR10, NDPDC can outperform [1] by over 20% accuracy, and NDPDC uses the lower budget.
>
> (2) In terms of LDPDC, according to our reproduced results (with our settings and limited information), LDPDC performs better than [1]. Specifically, We set $\sigma_X=\sigma_Y=1$, $L=50$, and $M=50$ (number of samples per class). The result we got for [1] is only 10.22\% $\pm$ 1.23\% on CIFAR10 (please check the probable reasons in our previous response). According to the results reported in [1], LDPDC uses much lower privacy budgets to achieve comparable performance to [1].
>
> **All in all, the improvement of our paper over [1] is obvious.**
>
> If our response addresses your major concern, please consider raising the score. Thank you very much for your time to review this paper!
>
> [1] Lee, Kangwook, et al. "Synthesizing differentially private datasets using random mixing." 2019 IEEE International Symposium on Information Theory (ISIT). IEEE, 2019.

---

> ### Author Response · Authors · 2022-11-21
> **Thank you for your valuable comments and recommendations**
>
> Dear Reviewer Spfc,
>
> We sincerely thank you for your time and efforts to review this paper. We greatly appreciate your valuable comments and recommendations, and we also appreciate the work done by [1]. So we follow your recommendations to further add discussion about LDPDC and [1] and fix the clarity issues in the revision.
>
> Last but not least, we thank you for your time to participate in the discussion. We also thank you for checking our response carefully and raising the score.
>
> Best,
>
> Authors
>
> [1] Lee, Kangwook, et al. "Synthesizing differentially private datasets using random mixing." 2019 IEEE International Symposium on Information Theory (ISIT). IEEE, 2019.

---

### Official Review · Reviewer_kpgg · 2022-10-26

**Confidence:** 3
**Clarity, Quality, Novelty And Reproducibility:** Paper was easy to read
**Correctness:** 3
**Technical Novelty And Significance:** 3
**Empirical Novelty And Significance:** 3
**Recommendation:** 6

**Strength And Weaknesses:**

Paper proposes a differentially private version of dataset condensation and proposes two DP methods, linear and non-linear differentially private dataset condensation. Proposed methods claim to avoid the pitfall of [1], and provide formal privacy guarantees.

Empirical evaluation shows that the proposed DP methods perform better than DP-sinkhorn and DP-MERF, and provide tighter privacy guarantees.

Overall, I like the paper. I have not checked the proofs in detail, assuming they go through, the proposed methods are potentially useful. I think a brief discussion on the utility and use of DP data condensation methods vs general DP data generation would be useful.

**Summary Of The Paper:**

Paper proposes differentially private dataset condensation, and improvement over the ICML'22 work [1], and shows that the proposed method provides good utility.

[1] Dong, T., Zhao, B., & Lyu, L. (2022). Privacy for Free: How does Dataset Condensation Help Privacy?. arXiv preprint arXiv:2206.00240.

**Summary Of The Review:**

I like the proposed method, and assuming the proofs are solid, I have no problem accepting it to the conference.

---

> ### Author Response · Authors · 2022-11-07
> **Thank you for your valuable review.**
>
> We sincerely thank you for your valuable time to review this paper, and we also thank you for your valuable comments.
>
> > Q1: A brief discussion on the utility and use of DP data condensation methods vs general DP data generation would be useful.
>
> A: We appreciate your insightful suggestion that can help us improve this paper.
> General DP data generation methods typically target at training generative models to generate new synthetic data samples, while DP dataset condensation methods aim to condense the original dataset into a small synthetic dataset and maintain the data utility for training models [1, 2, 3]. **Thus, DP data generation methods may be more useful for generating new data samples, but DP dataset condensation is more useful for data-efficient learning [1, 2, 3].** This is because DP dataset condensation significantly reduces the cost of data storage and model training, and the models trained on a small amount of data produced by DP dataset condensation methods have better utility than the models trained on a small amount of data generated by DP generative methods.
>
> A useful and practical application of DP dataset condensation methods is: Data owners (with privacy concerns) could condense their datasets into small synthetic datasets by DP dataset condensation methods. With DP protection, the data owners would feel more comfortable to share the synthetic datasets with other users or devices. Even if some users or devices do not have much computational resource or storage, they could store the small synthetic datasets and train models with not bad utility. The data owners could also share their data with trusted entities such as governments, hospitals, or banks. To mitigate the data owners' concerns, the trusted entities can execute DP dataset condensation methods on the collected data and share the small synthetic datasets with other users or devices. In this application, DP dataset condensation is a better choice than general DP data generation methods since the models trained on the small synthetic datasets produced by DP dataset condensation methods have better utility than the models trained on the data generated by DP generation methods.
>
> We will add the above discussion in the revision, and we believe that we could see more study and applications of DP dataset condensation in the future.
>
> Thanks again for your valuable comments. Please let us know if you have any other comments or suggestions.
>
>
> [1] Bo Zhao, Konda Reddy Mopuri, and Hakan Bilen. Dataset condensation with gradient matching. ICLR, 1(2):3, 2021.
>
> [2] Bo Zhao and Hakan Bilen. Dataset condensation with distribution matching. CoRR, abs/2110.04181, 2021
>
> [3] Bo Zhao and Hakan Bilen. Dataset condensation with differentiable siamese augmentation. In International Conference on Machine Learning, pp. 12674–12685. PMLR, 2021

---

### Official Review · Reviewer_gV1f · 2022-10-28

**Confidence:** 2
**Correctness:** 4
**Technical Novelty And Significance:** 3
**Empirical Novelty And Significance:** 2
**Recommendation:** 6

**Clarity, Quality, Novelty And Reproducibility:**

I'm confused about Algorithm 2. I think the presentation might need to be improved to clarify things. And since I'm not able to follow the algorithm, I'm not quite sure about the correctness of the privacy analysis.
- S seems to be a set but then we have S = S - eta*gradient of l. How does that work? Is S like a concatenation of vectors?
- What is the gradient with respect to? Is theta some learnable parameter?
- I think it would be better if you write down the gradient explicitly.
- What does it mean by "initialize S ... with random noise"?

Some other comments:
- In Table 1, why don't we compare different methods under the same privacy budget?
- I'm confused about the formulation in (1). Wouldn't it be optimal if I choose S = T? I feel like there need to be some term related to how condense / private S is, right?

--- update ---

The authors' response and revision has cleared my questions.

**Strength And Weaknesses:**

Strength: The idea seems interesting and the empirical results seem good.

Weakness: I'm quite confused about how the algorithm works.

**Summary Of The Paper:**

The paper proposes a data condensation algorithm that guarantees differential privacy, and demonstrated empirically that the proposed algorithm works better than existing data condensation algorithms.

**Summary Of The Review:**

The general idea seems interesting, but I wasn't able to understand Algorithm 2 to judge the correctness of privacy.

---

> ### Author Response · Authors · 2022-11-07
> **A detailed response to address your concerns and questions. (We eagerly look forward to your valuable feedback).**
>
> We appreciate your valuable time to review this paper. We provide a detailed response to your questions and concerns below. **We eagerly look forward to your valuable feedback on our response.**
>
> > Q1: How does $\mathcal{S} = \mathcal{S} - \eta\nabla_{\mathcal{S}}\ell$ work? Is $\mathcal{S}$ like a concatenation of vectors?
>
> A: We follow Algorithm 1 in [1] to write the optimization step for $\mathcal{S}$ as $\mathcal{S} = \mathcal{S} - \eta\nabla_{\mathcal{S}}\ell$. **In practical implementation**, $\mathcal{S}$ is a tensor variable with size $[N,$ data_shape$]$ (e.g., $[N, 3, 32, 32]$ on CIFAR10), where $N$ is the size of the synthetic dataset. So we could compute the gradient of the loss $\ell$ with respect to the tensor variable $\mathcal{S}$ and update $\mathcal{S}$. We note that this optimization step on $\mathcal{S}$ is similar to the optimization step in [1, 2, 3], and Algorithm 2 (NDPDC) guarantees RDP (for each iteration) by adding noise to the sum of the clipped original data representations before this optimization step.
>
> > Q2: What is the gradient with respect to? Is $\theta$ some learnable parameter?
>
> A: The gradient $\nabla_{\mathcal{S}}\ell$ is with respect to the tensor variable $\mathcal{S}$ in practical implementation, or we can say it is with respect to the synthetic data samples. $\theta$ is randomly sampled and fixed in each iteration, which is the same as [1].
>
> > Q3: It would be better if you write down the gradient explicitly.
>
> A: Thank you for this suggestion to improve clarity. We could rewrite $\mathcal{S} = \mathcal{S} - \eta\nabla_{\mathcal{S}}\ell$ as $\mathbf{\mathit{s}}^c_j = \mathbf{\mathit{s}}^c_j - \eta\nabla_{\mathbf{\mathit{s}}^c_j }\ell ~\forall \mathbf{\mathit{s}}^c_j \in \mathcal{S}$. Those two expressions are equivalent, and you can consider $\mathcal{S}$ as a batch of $\mathbf{\mathit{s}}^c_j$. We have included the expression $\mathbf{\mathit{s}}^c_j = \mathbf{\mathit{s}}^c_j - \eta\nabla_{\mathbf{\mathit{s}}^c_j }\ell ~\forall \mathbf{\mathit{s}}^c_j \in \mathcal{S}$ behind $\mathcal{S} = \mathcal{S} - \eta\nabla_{\mathcal{S}}\ell$ in Algorithm 2 in the revision.
>
> > Q4: What does it mean by "initialize $\mathcal{S}$ ... with random noise"?
>
> A: It means that Initializing the tensor variable $\mathcal{S}$ with random Gaussian noise, **which is the same as [1, 2, 3]**. The Pytorch code is like "image_syn = torch.randn(size=(num_classes*args.ipc, channel, im_size[0], im_size[1]), dtype=torch.float, requires_grad=True, device=args.device)". [1, 2, 3] also have another initialization option called initializing $\mathcal{S}$ with real images. We do **not** use the real image initialization option since we need DP protection.
>
> > Q5: In Table 1, why don't we compare different methods under the same privacy budget?
>
> A: This is because different papers' default settings use different budgets, e.g., DP-Sinkhorn uses
> $\epsilon=10$ but DP-MERF uses $\epsilon=1$. We conduct experiments under $\epsilon=1$ and provide results in the following table. For all the methods, we use number_classes * 50 synthetic samples ($M=50$) for training the models, and test the models on the real testing datasets. It is clear that NDPDC has much better performance than the other methods.
>
> | $\epsilon=1$ |        MNIST       |        FMNIST      |       CIFAR10      | CelebA (2 classes) |
> |:------------:|:------------------:|:------------------:|:------------------:|:------------------:|
> |    LDPDC     | 85.80\%$\pm$0.39\% | 63.64\%$\pm$0.76\% | 25.46\%$\pm$0.83\% | 69.64\%$\pm$1.01\% |
> |    NDPDC     | 95.32\%$\pm$0.29\% | 78.79\%$\pm$0.37\% | 42.40\%$\pm$0.86\% | 81.47\%$\pm$0.80\% |
> | DP Sinkhorn  | 55.43\%$\pm$1.54\% | 43.22\%$\pm$1.40\% | 12.62\%$\pm$0.27\% | 64.02\%$\pm$0.48\% |
> |   DP MERF    | 84.81\%$\pm$2.04\% | 63.05\%$\pm$2.05\% | 17.10\%$\pm$0.78\% | 69.26\%$\pm$0.90\% |
>
>
> > Q6: Wouldn't it be optimal if I choose $\mathcal{S} = \mathcal{T}$ for the formulation in (1)?
>
> A: For dataset condensation, the size of $\mathcal{S}$ should be much smaller than the size of $\mathcal{T}$ ($|\mathcal{S}| \ll |\mathcal{T}$|). Suppose $|\mathcal{S}|=500$ and $|\mathcal{T}|=50000$, then $\mathcal{S} = \mathcal{T}$ is not a feasible solution since it is impossible to achieve $\mathcal{S} = \mathcal{T}$. In the revision, we add $|\mathcal{S}| \ll |\mathcal{T}|$ in the formulation (1) to improve the clarity.
>
> **If we have addressed your major concerns, we hope you will consider raising the score. We eagerly look forward to your valuable feedback. Thanks again for your valuable review.**
>
>
> Reference
>
> [1] Bo Zhao and Hakan Bilen. Dataset condensation with distribution matching. CoRR, abs/2110.04181, 2021
>
> [2] Bo Zhao, Konda Reddy Mopuri, and Hakan Bilen. Dataset condensation with gradient matching. ICLR, 1(2):3, 2021.
>
> [3] Bo Zhao and Hakan Bilen. Dataset condensation with differentiable siamese augmentation. In International Conference on Machine Learning, pp. 12674–12685. PMLR, 2021

---

> > ### Comment · Reviewer_gV1f · 2022-12-11
> > **Thanks!**
> >
> > Thanks for the detailed response and sorry for my misunderstanding before about Alg 2.
> > I have updated my review. Thanks the authors for the response and revision!
> >
> > A minor point: when I said "write down the gradient explicitly", I was a bit confused about the privacy computation, i.e., which quantity would guarantee privacy for us. So what I meant is more like expanding the gradient to a formula without $\nabla$, and stating which part (I guess the N + G(D_c) part) guarantees privacy.

---

> > > ### Author Response · Authors · 2022-12-11
> > > **Thank you for your valuable comments**
> > >
> > > We sincerely thank you for your valuable comments.
> > >
> > > > A minor point: when I said "write down the gradient explicitly", I was a bit confused about the privacy computation, i.e., which quantity would guarantee privacy for us. So what I meant is more like expanding the gradient to a formula without
> > > ∇, and stating which part (I guess the N + G(D_c) part) guarantees privacy.
> > >
> > > A: This is a good point. According to the chain rule, the derivative of the loss w.r.t. a synthetic data sample $\mathbf{\mathit{s}}^c_j$ is like
> > >
> > > $$\frac{2L}{M}(\frac{L}{M}\sum_{j=1}^{M}\\mathbf{\mathit{\tilde{r}}}(\mathbf{\mathit{s}}^c_j) - (\mathcal{N}(\mathbf{\mathit{0}}, \sigma^2 \mathbf{\mathit{I}}) + \sum_{i=1}^{|D_c|} \mathbf{\mathit{\tilde{r}}}(\mathbf{\mathit{x}}_i^c)))^T\frac{\partial\mathbf{\mathit{\tilde{r}}}(\mathbf{\mathit{s}}^c_j)}{\partial\mathbf{\mathit{s}}^c_j}$$
> > >
> > > The above derivate is $1\times d$ format, where d is the dimension of $\mathbf{\mathit{s}}^c_j$. The transpose of the above formula is the $d\times 1$ format derivate. $\frac{\partial\mathbf{\mathit{\tilde{r}}}(\mathbf{\mathit{s}}^c_j)}{\partial\mathbf{\mathit{s}}^c_j}$ is a $d_r \times d$ matrix, where $d_r$ is the dimension of the representation $\mathbf{\mathit{\tilde{r}}}(\mathbf{\mathit{s}}^c_j)$. This matrix $\frac{\partial\mathbf{\mathit{\tilde{r}}}(\mathbf{\mathit{s}}^c_j)}{\partial\mathbf{\mathit{s}}^c_j}$ does not leak additional private information about real data $\mathbf{\mathit{x}}$ in one iteration (We only consider one iteration and use composition law). The vector before the matrix, i.e., $(\frac{L}{M}\sum_{j=1}^{M}\\mathbf{\mathit{\tilde{r}}}(\mathbf{\mathit{s}}^c_j) - (\mathcal{N}(\mathbf{\mathit{0}}, \sigma^2 \mathbf{\mathit{I}}) + \sum_{i=1}^{|D_c|} \mathbf{\mathit{\tilde{r}}}(\mathbf{\mathit{x}}_i^c)))^T$, already guarantees RDP. According to the postprocessing property, the derivate guarantees RDP.
> > >
> > > Compared to expanding the derivate, a more intuitive way to understand the privacy guarantee may be directly considering the postprocessing property after the adding noise step.
> > >
> > >
> > > **We really appreciate your valuable time to review this paper and check our response. Please let us know if you have any other question.**

---

> ### Author Response · Authors · 2022-11-16
> **Could you please check our response? Thank you in advance for your kind cooperation!**
>
> Dear Reviewer gV1f,
>
> Since there is little time left in the first discussion period, we would greatly appreciate it if you could provide some feedback on our response and revision soon. We believe we have addressed your major concern.
>
> > How does $\mathcal{S} = \mathcal{S} - \eta\nabla_{\mathcal{S}}\ell$ (optimization step on $\mathcal{S}$) work? (**Please refer to our answers to the other questions and concerns in the previous response**).
>
> We follow Algorithm 1 in [1] to write the optimization step for $\mathcal{S}$ as $\mathcal{S} = \mathcal{S} - \eta\nabla_{\mathcal{S}}\ell$. **In practical implementation**, $\mathcal{S}$ is a tensor variable with size $[N,$ data_shape$]$ (e.g., $[N, 3, 32, 32]$ on CIFAR10), where $N$ is the size of the synthetic dataset. So we could compute the gradient of the loss $\ell$ with respect to the tensor variable $\mathcal{S}$ and update it. We note that this optimization step on $\mathcal{S}$ is similar to the optimization step in [1, 2, 3], and Algorithm 2 (NDPDC) already guarantees RDP (for each iteration) by adding noise to the sum of the clipped original data representations before this optimization step.
>
> **We have provided more details in our previous response.**
>
> In the revision, we add explanation in Page 6 (highlighted by the blue color). We also provide a more clear version of the formula in Algorithm 2 (NDPDC).
>
> **If our response addresses your major concern, please consider raising the score. Thank you very much for your time to review this paper!**
>
>
>
> [1] Bo Zhao and Hakan Bilen. Dataset condensation with distribution matching. CoRR, abs/2110.04181, 2021
>
> [2] Bo Zhao, Konda Reddy Mopuri, and Hakan Bilen. Dataset condensation with gradient matching. ICLR, 1(2):3, 2021.
>
> [3] Bo Zhao and Hakan Bilen. Dataset condensation with differentiable siamese augmentation. In International Conference on Machine Learning, pp. 12674–12685. PMLR, 2021

---

> ### Author Response · Authors · 2022-11-21
> **Follow Up with Reviewer gV1f**
>
> Dear Reviewer gV1f,
>
> Since we already had a fruitful discussion with Reviewer Spfc and addressed Reviewer Spfc’s major concern, we are now eagerly looking forward to your valuable feedback.
>
> Does our response address your major concern? If so, we hope you will consider raising the score. Please kindly let us know if you have further questions, and we will be happy to discuss with you. We sincerely thank you for your efforts in reviewing this paper!
>
> Best,
>
> Authors

---

### Author Response · Authors · 2022-11-16
**Summary of Revision**

We thank all the reviewers for their valuable time to review this paper. Here is a summary of the modifications in the revision. More details could be found in our responses to the reviewers.

> Major Concerns:

> Reviewer gV1f: How does $\mathcal{S} = \mathcal{S} - \eta\nabla_{\mathcal{S}}\ell$ (optimization step on $\mathcal{S}$) work?

We add explanation in Page 6 (highlighted by the blue color). We follow Algorithm 1 in [1] to write the optimization step for $\mathcal{S}$ as $\mathcal{S} = \mathcal{S} - \eta\nabla_{\mathcal{S}}\ell$. **In practical implementation**, $\mathcal{S}$ is a tensor variable with size $[N,$ data_shape$]$ (e.g., $[N, 3, 32, 32]$ on CIFAR10), where $N$ is the size of the synthetic dataset. So we could compute the gradient of the loss $\ell$ with respect to the tensor variable $\mathcal{S}$ and update it. Besides, we provide a more clear expression of this optimization step, i.e., $\mathbf{\mathit{s}}^c_j = \mathbf{\mathit{s}}^c_j - \eta\nabla_{\mathbf{\mathit{s}}^c_j }\ell ~\forall \mathbf{\mathit{s}}^c_j \in \mathcal{S}$, behind $\mathcal{S} = \mathcal{S} - \eta\nabla_{\mathcal{S}}\ell$ in Algorithm 2 in the revision. We note that this optimization step on $\mathcal{S}$ is similar to the optimization step in [2, 3, 4], and Algorithm 2 (NDPDC) already guarantees RDP (for each iteration) by adding noise to the sum of the clipped original data representations before this optimization step.

> Reviewer Spfc: It is questionable how much improvement LDPDC and NDPDC have by comparing them with [1]?

According to the results reported in [1], NDPDC outperforms [1] by 13\%-24\% in testing accuracy on MNIST and CIFAR10. This is because [1] is a linear algorithm, while NDPDC is a nonlinear algorithm that is more suitable for nonlinear problems such as image recognition. In the revision, we clarify the improvement of NDPDC over [1] in Section 4.2 (also highlighted by the blue color). We also add discussion on LDPDC and [1] in Appendix F.

> Other Issues:

> Reviewer gV1f and Reviewer Spfc: Comparison under the same budget?

This is because different papers' default settings use different budgets, e.g., DP-Sinkhorn uses
$\epsilon=10$ but DP-MERF uses $\epsilon=1$. In the revision, We provide the results of LDPDC, NDPDC, DP Sinkhorn, and DP-MERF under $\epsilon=1$ in Table 2, which clearly shows that NDPDC has better performance than the other methods.

> Reviewer kpgg: Discussion on DP dataset condensation vs. general DP data generation?

In the revision, we provide more discussion in Appendix F.

> Reviewer Spfc: Clarity issues

We fix the clarity issues in the revision.


[1] Lee, Kangwook, et al. "Synthesizing differentially private datasets using random mixing." 2019 IEEE International Symposium on Information Theory (ISIT). IEEE, 2019.

[2] Bo Zhao and Hakan Bilen. Dataset condensation with distribution matching. CoRR, abs/2110.04181, 2021

[3] Bo Zhao, Konda Reddy Mopuri, and Hakan Bilen. Dataset condensation with gradient matching. ICLR, 1(2):3, 2021.

[4] Bo Zhao and Hakan Bilen. Dataset condensation with differentiable siamese augmentation. In International Conference on Machine Learning, pp. 12674–12685. PMLR, 2021

---

> ### Author Response · Authors · 2022-11-17
> **Further changes in the revision to address Reviewer Spfc's concern**
>
> We thank Reviewer Spfc for Reviewer Spfc's valuable time to review this paper!
>
> In the revision, we further add discussion about LDPDC and [1] in Appendix F.
>
> > Comparison between LDPDC and [1]
>
> In the revision, we add discussion on the three major differences between LDPDC and [1]:
>
> (1) LDPDC does not need to randomize the labels (thanks to the parallel composition law), but DPMix [1] needs to randomize the labels. **Adding noise to the labels may hurt the model performance.**
>
> (2) LDPDC adds noise to the sum of samples and divides it by the fixed group size $L$, while [1] directly adds noise to the mean of the samples. For LDPDC, dividing the noise + sum by the group size $L$ could help dilute the negative effects of random noise on model performance. **We find that, even for LDPDC, if we directly add noise to mean of samples like [1], the performance of LDPDC will also become much worse.**
>
> (3) DPMix uses sampling without replacement, while LDPDC uses Poisson sampling, which is the standard sampling method adopted in the state-of-the-art Pytorch library for differentially private deep learning [2].
>
> In terms of performance, we have reproduced [1] with our settings and limited information in the published version of [1]. If we set $\sigma_X=\sigma_Y=1$, $L=50$, and $M=50$, the result we got for [1] is only 10.22\% $\pm$ 1.23\% on CIFAR10. **So according to our reproduced results, LDPDC has better performance than [1]. Also, according to the results reported in [1], LDPDC uses lower DP budgets to achieve comparable performance to [1].**
>
> > Novelty of LDPDC
>
> We believe that NDPDC and the extensive theoretical and empirical analysis are the most important contributions of this paper. The improvement of NDPDC over [1] is significant. We include LDPDC in this paper mainly because it is simple to use (with our Python code) and has comparable performance to recent DP generative methods. So from our perspective, the novelty of LDPDC (compared to [1]) should not be used as the only major reason to reject this paper.
>
> **We thank all the reviewers for their valuable time to review this paper!**
>
>
> [1] Lee, Kangwook, et al. "Synthesizing differentially private datasets using random mixing." 2019 IEEE International Symposium on Information Theory (ISIT). IEEE, 2019.
>
> [2] Yousefpour, Ashkan, et al. "Opacus: User-friendly differential privacy library in PyTorch." arXiv preprint arXiv:2109.12298 (2021).

---

> > ### Author Response · Authors · 2022-11-18
> > **More modifications in the revision based on Reviewer Spfc's recommendations (We thank Reviewer Spfc for the valuable comments.)**
> >
> > We follow Reviewer Spfc's recommendations to further modify the revision for discussing LDPDC and [1] in Appendix F and fixing the clarification issues.
> >
> > We thank Reviewer Spfc for the valuable comments that help us improve the paper. We also thank Reviewer Spfc for participating in the discussion and raising the score to 6.
> >
> > We thank all the reviewers again for their valuable time to review this paper!
> >
> > [1] Lee, Kangwook, et al. "Synthesizing differentially private datasets using random mixing." 2019 IEEE International Symposium on Information Theory (ISIT). IEEE, 2019.

---

### Decision · Program_Chairs · 2023-01-20

**Decision:**

Reject

**Justification For Why Not Higher Score:**

Some lack of novelty over prior work and unclear relationship with it, not obvious what aspect of the new method results in better performance, critique of previous work is somewhat indirect.

**Justification For Why Not Lower Score:**

N/A

**Metareview: Summary, Strengths And Weaknesses:**

This paper resulted in some additional discussion between reviewers. One point brought up was that NDPDC is rather similar, conceptually, to DP-MERF. This implies the following:
- Both techniques in the paper seem very similar to prior works, with some slight modification.
- NDPDC is conceptually similar to DP-MERF, but seems to perform empirically better. This comparison may need to be done better, to identify what difference between the two methods really results in better utility.

Another issue raised was that this paper seems to dodge the fundamental critiques against the paper of Dong et al. For example, the example of Assumption 3.1 was not only the details of the fact that this would be implausible in, e.g., non-stochastic and full batch gradient descent settings, but even the "correct" Assumption 3.2 seems too strong to assume (the authors of the submission mention that it may be true in some asymptotic sense for some particular setting, but is unclear whether it's something that could hold more broadly). Furthermore, the understanding is that the proof of Proposition 4.10 is incorrect even for the linear case, whereas this paper shows that it is not true for the non-linear case. In these senses, the paper seems to be "weaselly" about the issues with the paper of Dong et al., getting lost among the trees inside the broader forest.

For these reasons, it was felt that the paper may benefit from further revision, to better clarify the contribution and differences with prior work, as well as being more direct with the shortcomings of the prior work that the authors critique.

Orthogonal to the decision, the authors may also wish to note and reference the recent (after the ICLR submission deadline) paper by Carlini, Feldman, and Nasr (https://arxiv.org/abs/2209.14987), which seems to fill the same narrow niche of critiquing this particular paper.